# Implementation of the NHS England Lung Cancer Screening Programme over 5 years

Richard W. Lee [1,2,3,4,60] ✉, Arjun Nair[5,60], Haval Balata[6,7], Charlotte Graham[4], Craig Parylo [8], Jessica Abell [4], Michael Woodall[8], Michael Lawrie [9], Sally Mouland[9], Kate Brain[10], Michelle Clark[11], Philip Crosbie[7], Anand Devaraj[3,12], Jesme Fox[13], Martin Grange[14], Sam M. Janes [15], Peter Johnson [4,16], Anne Mackie[17], Neal Navani [15], Samantha L. Quaife[18], Amelia Randle[19], Janette Rawlinson [20], Robert C. Rintoul [21,22], Liz Rochelle[17], Peter Sasieni [18], Matthew E. J. Callister [23,24] & David R. Baldwin [25] on behalf of UK Lung Cancer Screening Research Consortium*

Lung cancer screening with low-dose computed tomography has been proven to reduce lung-cancer-specific and all-cause mortality. The UK launched the NHS England Targeted Lung Health Check Programme in 2019, which has now become the national Lung Cancer Screening Programme, with full coverage expected by 2030. Here we present the progress and outcomes of the program. People aged 55–74 were offered low-dose computed tomography of the thorax if they had ever smoked and if risk thresholds, as determined by multivariable models, were met. Delivery of the program is through regionally federated clinical infrastructure and leadership, with national strategic, clinical and economic frameworks. The program has invited over two million people, with 7,193 lung cancers diagnosed—63.1% at tumor, node, metastasis stage 1 and 12.6% stage 2—to March 2025. This has increased the early-stage proportion of lung cancer in England over 5 years, particularly in socioeconomically deprived regions. The NHS England Programme exemplifies how large-scale implementation can be achieved at speed through centralized protocols and effective project management. The program has demonstrated feasibility and scalability in reaching high-risk and underserved populations, but needs to further address inequalities in participation. These findings support adoption of lung cancer screening across the UK and globally, and offer practical tools for international adaptation.

Lung cancer screening with low-dose computed tomography (LDCT) of the thorax has been shown by several randomized controlled trials to reduce lung-cancer-specific and all-cause mortality[1,2]. Globally, national-level progress in lung cancer screening varies from continued research, through evaluation of the clinical and cost-effectiveness of small pilot projects to full-scale implementation of national screening programs. The National Health Service (NHS) England Lung Cancer Screening Programme, known in its earlier, large-scale pilot phase as the Targeted Lung Health Check (TLHC) Programme, was launched in 2019, with the aim to improve the proportion of cancer detected at an early stage in England. NHS England had recently set a target of 75% of people with any cancer being diagnosed at stage 1 and 2 by 2028. The pilot was also expected to provide real-world evidence ahead of a revised evaluation by the UK National Screening Committee. The goal was to create a national strategic, clinical and economic framework, within which to achieve full-scale deployment. The pilot constituted an important

contribution to the national early cancer diagnosis agenda, while identifying challenges and solutions to clinical referral pathways for lung cancer and other clinically relevant findings. This would also inform commissioning budgets for a full rollout. The program adheres to a detailed national protocol[3] and quality assurance standard[4], through regionally federated clinical infrastructure and leadership.

The progress toward implementation began with the UK Lung Screen pilot randomized trial[5], which showed that lung cancer screening was feasible and able to achieve both a higher lung cancer detection rate and higher early-stage lung cancer proportion than the US National Lung Screening Trial, the first trial to confirm a mortality reduction[6]. Small nonrandomized pilot programs demonstrated similar results and, notably, very low levels of harm[7–11]. A large-scale pilot, the TLHC was approved by NHS England in 2019. In September 2022, lung cancer screening for high-risk individuals aged 55–74 years was formally recommended by the UK National Screening Committee after a detailed health economics evaluation[12]. The English government announced this in June 2023, with an acknowledgment that the annual cost would be £270 M per year at full rollout. On 1 February 2025, the TLHC was formally renamed as the NHS England Lung Cancer Screening Programme.

This Article reports on the progress and outcomes of the screening program to date, including the process of phased rollout, participant uptake, lung cancer detection and downstream management.

## Results

Table 1 summarizes the total number of invitations, lung health checks, baseline LDCTs and lung cancers diagnosed for both the whole program and initial phase. Whole-program data represent total activity in the program until March 2025 with over 4.5 times the number of LDCT and lung cancer diagnoses than the initial phase data.

### Whole-program lung screening performance metrics

In the whole-program data from April 2019 to March 2025, 2,510,092 participants had been invited for a baseline Lung Health Check (LHC) across all Cancer Alliances (Fig. 1, consort diagram). This amounts to an invitation (coverage) of 32.4% of the total estimated, potentially eligible population of people who have ever smoked, aged 55–74 (7,743,437). Of those invited, 49.0% (1,229,714) have undergone an LHC (uptake of offer), and of these 16.8% (206,516 of 1,229,714) were performed face to face, and 83.2% (1,023,198) were performed by telephone, followed by face-to-face confirmation for those participants who qualified for LDCT screening; 47.5% (584,095) of participants met the multivariable model risk threshold (Liverpool Lung Project$_{v2}$ (LLPv2) or Prostate Lung Colorectal Ovary$_{m2012}$ (PLCO$_{m2012}$) models) and 43.0% (528,686 of 1,229,714 LHCs) underwent a baseline LDCT, while 4.7% (27,236 of 584,095 who met the LHC risk threshold and qualified for LDCT) did not attend or canceled their LDCT and 2.3% (13,231 of 584,095) met the risk threshold but were ineligible for LDCT on the basis of the exclusion criteria. A total of 2.6% (14,942 of 584,095) had no record of having had a scan, with reason unknown. A total of 79,338 three-month surveillance scans were performed, 34,797 twelve-month scans and there were 136,194 scans undertaken at 24, 48 and 72 months. The latter includes participants in whom incident round scans would not have been performed by the time of data extraction, that is a participant may have had a baseline scan but not yet have reached the point of having a 24-month scan. Therefore, the number of surveillance and incident round scans may be underrepresented. Extended Data Figs. 1 and 2 summarize the program geography and proportional national rollout as a marker of coverage, according to Cancer Alliance. Figure 2a shows monthly (noncumulative) numbers of invites, LHCs completed and LDCT performed. Early efforts to initiate the program that were hindered by COVID-19 restrictions are notable until Spring 2021[13].

By March 2025, 7,193 lung cancers had been diagnosed (1.4% of baseline LDCT participants), 2,228 in the last year. The stage distribution was 63.1% stage 1, 12.6%, 12.6% and 8.8 % stage 2, 3 and 4,

respectively, with 2.8% stage not specified (Fig. 1). National Cancer Registration Data (NCRD) show that lung cancer early stage detection rates across the UK have increased steadily since the pandemic and are now well above pre-pandemic levels. Furthermore, since the lung screening program started, the proportion of stages 1 and 2 lung cancers in the most deprived socioeconomic quintile has increased from the lowest to the highest quintile (Fig. 2b). This has not been reported in other cancers.

### Initial-phase participant record-level demographic analysis

Of 582,700 people in the initial phase who were eligible for an LHC, 216,985 (37.2%) attended. The proportion of different groups of the eligible population, participation rates, LDCT attendance and cancer diagnoses are presented in Extended Data Table 1, while the odds ratios (ORs) are presented in Fig. 3; 303,825 of 582,700 (52%) of the eligible population were male, and 275,335 of 582,700 (48%) were female. LHC uptake in men and women as a proportion of the eligible male and female populations, respectively, were equivalent (113,670 of 303,825 (37.4%) versus 103,310 of 275,335 (37.5%)), but fewer women underwent an LDCT scan as a proportion of those attending an LHC (48.4% females versus 56.7% males). Specifically, of those assessed as high risk, women were also less likely to attend an LDCT than men (OR = 0.87, P < 0.001).

Among those assessed as high risk, older individuals were more likely to undergo LDCT compared to the reference group aged 55–64 years (65–74 years, OR = 1.09, P < 0.001; 75+ years, OR = 1.39, P < 0.001).

Ethnicity data were not known for 32.6% of the 582,700 individuals eligible for an LHC because of incomplete primary care records. The further a participant progressed through the pathway, the more likely ethnicity status would be recorded. A smaller proportion of participants invited for an LHC in the 'other' ethnic group attended than those of the 'white' ethnic group (18,295 of 97,265, 18.8% versus 184,765 of 295,410, 62.5%; OR = 0.15, 95% CI = 0.14–0.15, P < 0.001). In the subset of people assessed as high risk at LHC, those from ethnic groups other than white were significantly less likely to attend LDCT compared to those from the white group (OR = 0.79, P < 0.001).

People in areas of least deprivation (quintile 5) had a higher LHC uptake than the most deprived (quintile 1) (15,050 of 36,195, 41.6% versus 85,285 of 256,965, 33.2%, OR = 1.29, 95% CI = 1.25–1.32, P < 0.001). However, 37.6% (5,665 of 15,050) of LHC participants from the least deprived areas underwent LDCT scanning versus 57.2% (48,760 of 85,285) people from quintile 1. After a high-risk assessment, those living in the most deprived quintile (quintile 1) were less likely to attend LDCT than those in quintiles 2 (OR = 1.05, P < 0.05), 3 (OR = 1.19, P < 0.001) and 4 (OR = 1.29, P < 0.001), but not significantly different to quintile 5 (OR = 1.02, P = 0.6).

### Lung cancer detection in the initial phase

By January 2023, 74,202 participants had undertaken a baseline LDCT (Table 2). Three-month and 12-month nodule surveillance LDCTs were completed in 9,995 (13.5%) and 6,689 (9.0%), respectively. Three-month and 12-month scan data are not mutually exclusive (nodule assessment often requires both time points) and scans were censored at March 2024. Cancers (censored at August 2023) diagnosed from the baseline, 3-month and 12-month time point scans were 890, 135 and 70 (74.4%, 11.3% 5.9%) of 1,196 cancers diagnosed in the initial-phase data at all time points. This equates to a cumulative cancer conversion rate of 1.2%, 1.4%, and 1.5%, respectively of 74,202 participants. Twenty-four-month scans (most would represent incident round scans, and a minority of nodule surveillance scans) were documented in 24,933 participants of whom 36 were diagnosed with cancer (3.0% of 1,196 screen-detected cancers across both rounds); 2,393 scans at 'other' (undefined) time points detected 65 cancers. The total cancer detection proportion was 1.6% across both rounds.

Initial-phase data restricted to a follow-up period of at least 185 days from LDCT were available for 53,430 people; 2.9% (1,565 of 53,430)

**Table 1 | Comparison of initial-phase and whole-program samples**

| | Whole-program data | | | Initial-phase data | | |
|---|---|---|---|---|---|---|
| | **Number** | **Data source** | **Time period** | **Number** | **Data available** | **Time period** |
| Eligible (England) | 7,743,437 | NHS England | April 2019 to March 2029 | 582,700 | Record level | April 2019 to March 2024 |
| Invited to a Lung Health Check (first invites) | 2,510,092 | Aggregate | April 2019 to March 2025 | 571,685 | Record level | April 2019 to March 2024 |
| Attended a Lung Health Check | 1,229,714 | Aggregate | April 2019 to March 2025 | 216,985 | Record level | April 2019 to March 2024 |
| Attended a baseline LDCT scan | 528,686 | Aggregate | April 2019 to March 2025 | 114,430 | Record level | April 2019 to March 2024 |
| Lung cancer diagnosed | 7,193 | Aggregate | April 2019 to March 2025 | 1,565 | Record level linked to NCRAS lung cancer activity | April 2019 to August 2023 |

Demographic breakdowns and outcomes after specific events (for example, lung cancer diagnosed following a baseline scan) are only available for the initial-phase sites. NCRAS, National Cancer Registration and Analysis Service.

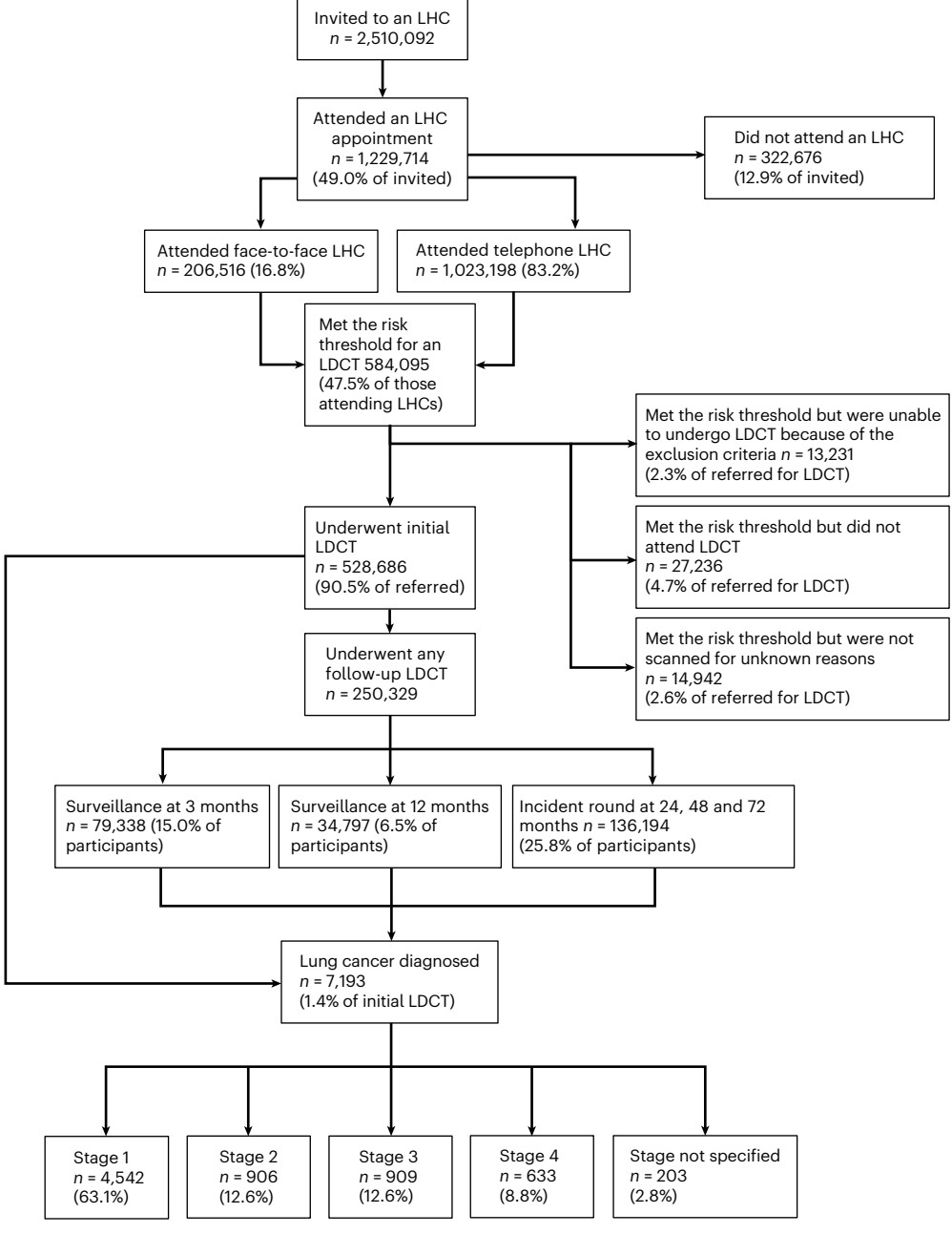

**Fig. 1 | Consort diagram.** Whole-program data to March 2025.

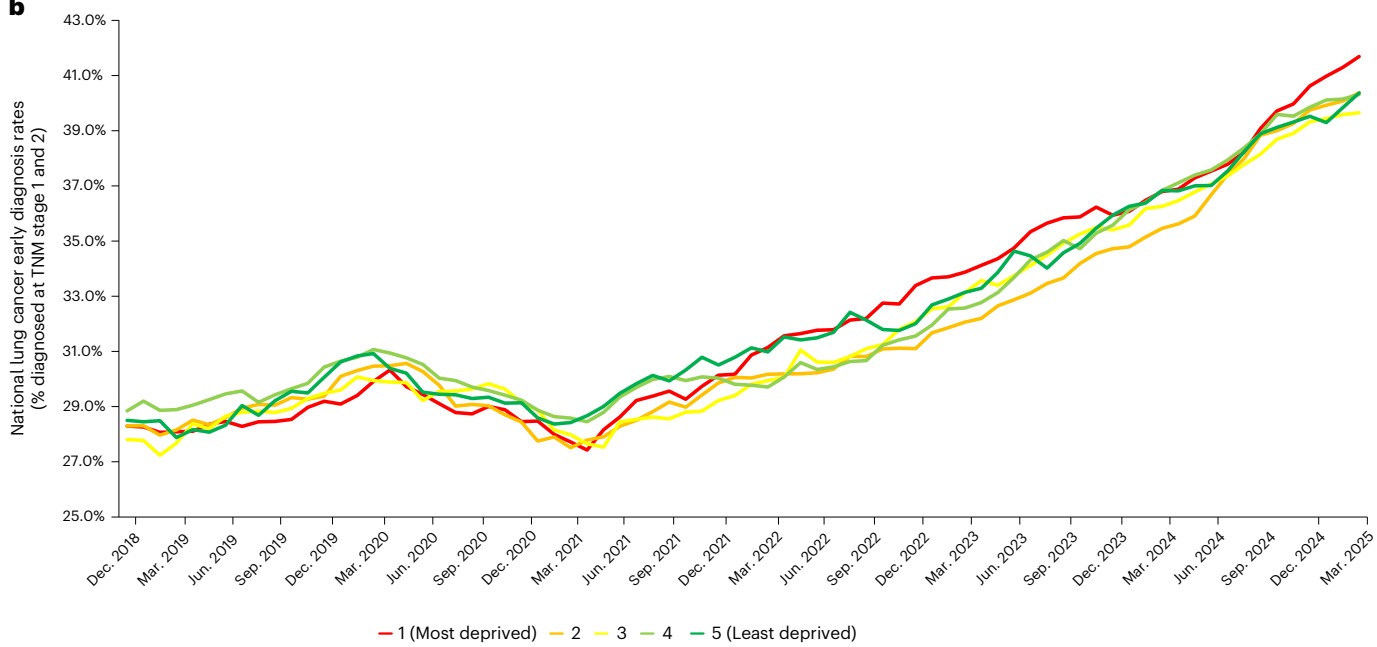

**Fig. 2 | Cancer screening and early diagnosis delivery metrics. a**, Lung cancer screening delivery trajectories. The number of first invites (people invited), LHCs and scans (across every scanning round) per month for the whole program, from May 2019 to March 2025, are shown. **b**, National lung cancer early diagnosis rates. Early diagnosis rates (% diagnosed at tumour node metastasis (TMN) stages 1 and 2), as a proportion of all lung cancer diagnoses, have increased for all deprivation quintiles after the pandemic, with the biggest change in stage (low to high rate) seen among those living in the most deprived areas. Source: NHS England Analysis of Rapid Registration Data.

of participants had a lung cancer diagnosis in the Cancer Outcomes and Services Data (COSD)-linked data within 185 days of LDCT. Cancer outcomes were censored in August 2023, representing 185 days after LDCT (including those for nodule surveillance). The objective of this was to ensure that all cancer diagnoses resulted from the LDCT. This time was chosen after an analysis confirming that this did not include cancers diagnosed that were not related to the program. This approach was only for the initial-phase data. In this subset, cancers diagnosed were

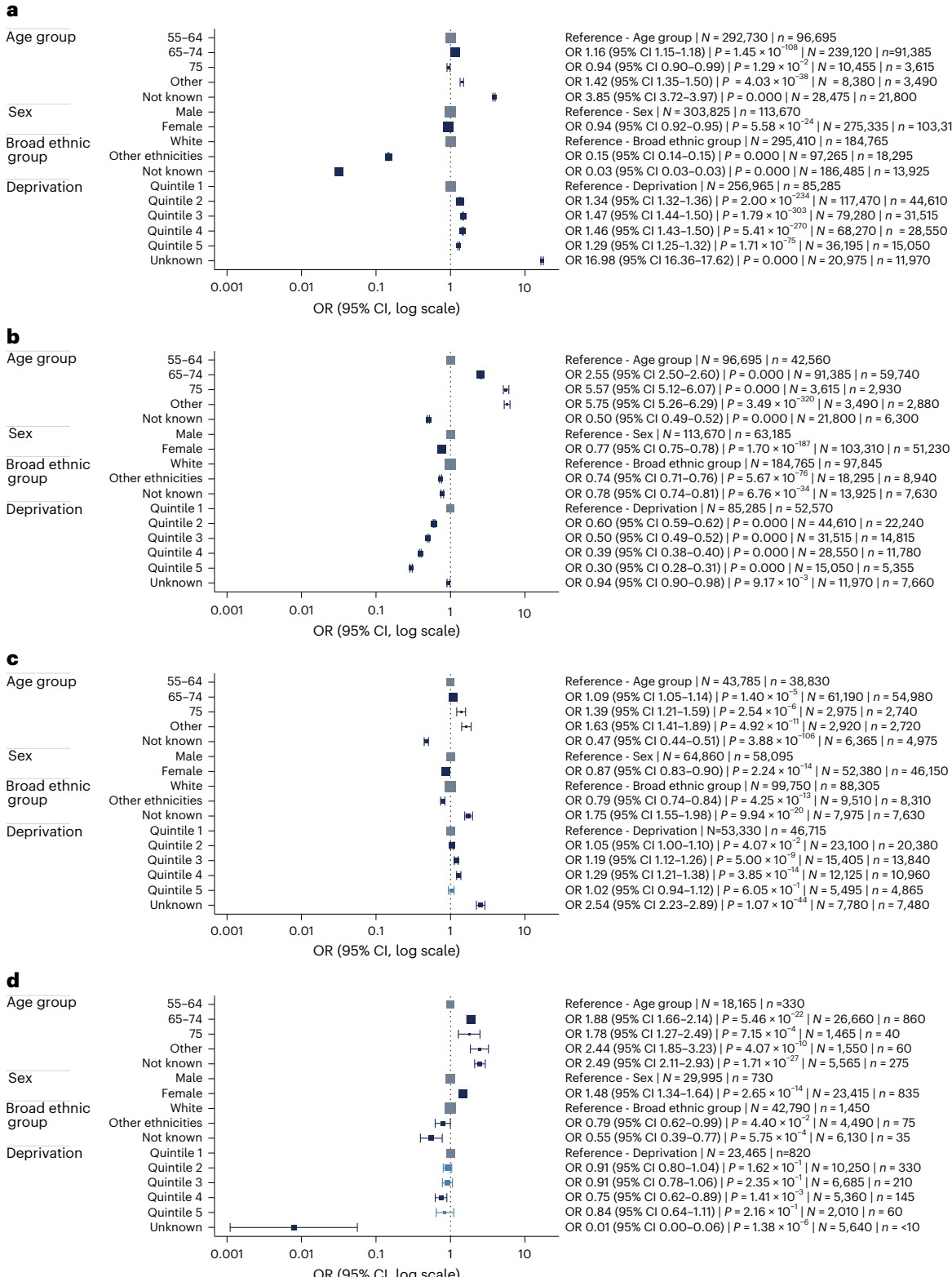

**Fig. 3 | Patient demographic determinants of lung screening engagement and outcomes.** OR plots of initial-phase data assessing the associations between demographic characteristics. **a**, LHC attendance, of those in the eligible population. **b**, Participants assessed as high risk. **c**, LDCT attendance of those attending an LHC and high risk. **d**, Lung cancers detected of those who attended LDCT with ≥185-day follow-up. ORs represent point estimates and the accompanying error bars show the 95% confidence interval (CI). The size of each square is proportional to N, meaning that larger squares correspond to a greater number of individuals included at the start of the regression. Estimates were obtained using multivariable logistic regression. P values were calculated using two-sided Wald tests based on the standard normal distribution, with no adjustment for multiple comparisons. N refers to the total number of individuals included at the start of the regression (for example, people in the eligible population) and n refers to the number who experienced the outcome of interest (for example, those who attended an LHC). Counts were rounded to the nearest multiple of five and values below ten were suppressed to protect confidentiality.

**Table 2 | Initial-phase LDCT scan timepoints and lung cancer detection (baseline scans before January 2023)**

| **(A) Baseline, Nodule Surveillance, and Incident round LDCT scans and lung cancers diagnosed** | | | | |
|---|---|---|---|---|
| LDCT scan time point | People | People diagnosed with screen-detected lung cancers | Proportion of screen-detected lung cancers (of 1,196 people) | Cumulative cancer detection rate (of 74,202 people) |
| Initial scan | 74,202 | 890 | 74.4% | 1.2% |
| 3-month follow-up scan | 9,995 | 135 | 11.3% | 1.4% |
| 12-month follow-up scan | 6,689 | 70 | 5.9% | 1.5% |
| 24-month follow-up scan | 24,933 | 36 | 3.0% | 1.5% |
| 48-month follow-up scan | 44 | 0 | 0.0% | 1.5% |
| Other scan | 2,393 | 65 | 5.4% | 1.6% |
| | **Total** | **1,196** | **100%** | **1.6%** |
| **(B) The number of people in the cohort by the year of their initial scan** | | | | |
| Year | | | People | |
| 2019–20 | | | 131 | |
| 2020–21 | | | 1,233 | |
| 2021–22 | | | 36,611 | |
| 2022–23 | | | 36,227 | |
| **Total** | | | **74,202** | |

Nodule Surveillance scans from initial-phase, participant-level data presented for people with a baseline scan before January 2023. All subsequent scans until March 2024, and corresponding lung cancer diagnoses and rates are available in NCRAS until August 2023. In both data, LDCT scan time points are not additive; many 3-month follow-up scans result in a further 12-month scan.

significantly higher in women than men (835 of 23,415, 3.6% in women, 730 of 29,995, 2.4% in men; OR = 1.48, 95% CI = 1.34–1.64, $P < 0.001$). Older (for example, 65–74 years) participants were also more likely to have a cancer diagnosis than younger (55–64 years) participants (OR = 1.88 CI = 1.66–2.14, $P < 0.001$) (Fig. 3 and Extended Data Table 1).

### Incidental findings

Assessment of non-lung-cancer LDCT scan findings was made in 114,430 participants selected from the initial phase who underwent baseline LDCT scanning (Table 3); 54,695 (47.8%) had documented coronary artery calcification (severity grading was not collected), 36,745 (32.1%) had aortic valve calcification and 13,830 (12.1%) had emphysema (radiologically moderate or severe). Of note, 525 (0.46%) other (non-lung) cancers were diagnosed.

## Discussion

The NHS England Lung Cancer Screening Programme (formerly TLHC) is a large, publicly funded national lung screening program that has scaled up at a fast pace by adopting a federated delivery model using a single mandated protocol and quality assurance standard. Both the delivery model and the program data provide a rich resource of real-world evidence for similar large-scale national programs that are at the stage of implementation. The scale of the program, with almost one-third of England's total estimated eligible population having received invitations, demonstrates that implementation at scale is feasible. The whole-program results imply that the protocol is working, with 1.4% of participants diagnosed with lung cancer, of which 76% were at stages 1 and 2. Although the proportion of participants with lung cancer is lower than that observed in the British pilots, which was an average of 2% at the baseline round, it is in keeping with, or above, the proportion seen in the large randomized trials[1,5,8–11,14].

The program has already shown an impact on the early-stage proportion in national lung cancer registry data, which has risen as screen-detected lung cancer has risen. No similar trend has been seen in other cancers in England. This demonstrates the impact that can be expected and was highlighted in an independent review of the NHS in England in 2024[15]. The latest UK National Lung Cancer Audit reports increases in lung cancer incidence, stage 1 rates and surgical resection rates[16]. While data currently only show an increase in stages 1 and 2, the similar stage distribution to randomized trials showing a reduction in mortality suggests that a mortality reduction can be expected.

Most (83.2%) LHCs were delivered by telephone, an innovation adapted into the Standard Protocol in response to the COVID-19 pandemic, when it was found to be feasible and efficient. Although following the Standard Protocol and structural quality assurance standard is mandated, innovations can be proposed for review by the national operations team and clinical advisory committee. This flexibility is important to facilitate iterative improvements identified by the separate sites, while maintaining uniformity, which is felt to be a strength of the program.

The uptake of the offer for an LHC, 49.0% overall, is lower than in other screening programs but screening was initially targeted in the areas of highest socioeconomic deprivation, where uptake is lower in all screening programs. Our analysis of the initial-phase data confirmed that participants in the most deprived socioeconomic group and in non-white communities were less likely to respond to the invite. However, uptake is improving as the program progresses, with the latest data indicating over 60% (Fig. 2), comparing very favorably to other international experience in lung cancer screening. This may reflect the strong public engagement and careful steps taken to ensure that effective invitation methods were followed based on evidence from lung cancer and other screening programs[17]. Despite these encouraging findings, there should be concern about the people who choose not to respond. Even if it is assumed that the same proportion would be eligible for screening, this means that half of the population do not currently have the chance to benefit. As socioeconomic disadvantage is associated both with lower participation and lung cancer, it could be that an even greater proportion of those who benefit are being missed. In-service evaluation of changes to the program are important. An example is the recent change to the name of the program from 'targeted lung health check' to 'lung cancer screening'. Before this, NHS England undertook a behavioral science-led assessment of a number of new names for both the program and the assessment. Surprisingly, participants preferred lung cancer screening for the program name because it was less ambiguous than 'targeted lung health check' but preferred 'lung health check' for the risk assessment by telephone or face to face. Further patient experience data are available from NHS England[18].

**Table 3 | Number of people with incidental findings reported (n=77,185) in the initial-phase projects (April 2019–March 2024), as a proportion of total who had an LDCT (N=114,430)**

| Incidental finding | Number of people with incidental finding recorded | Percent of people with incidental finding/ total who had an LDCT (N=114,430) |
|---|---|---|
| Coronary calcification | 54,695 | 47.80% |
| Aortic valve calcification | 36,745 | 32.11% |
| Emphysema (moderate or severe) | 13,830 | 12.09% |
| Thoracic aortic aneurysm | 1,970 | 1.72% |
| Interstitial lung abnormalities | 1,530 | 1.34% |
| Liver or spleen lesions | 1,115 | 0.97% |
| Renal lesions | 1,005 | 0.88% |
| Adrenal lesions | 930 | 0.81% |
| Bronchiectasis | 825 | 0.72% |
| Bone abnormalities | 725 | 0.63% |
| Osteoporosis | 635 | 0.56% |
| Pleural effusion thickening | 630 | 0.55% |
| Respiratory bronchiolitis | 540 | 0.47% |
| Consolidation | 530 | 0.46% |
| Other cancers | 525 | 0.46% |
| Mediastinal mass | 465 | 0.41% |
| Suspicious breast lesion | 430 | 0.37% |
| Fractures with no trauma history | 205 | 0.18% |
| Thyroid lesion | 200 | 0.17% |
| Abdominal aortic aneurysm | 65 | 0.06% |
| Tuberculosis | <10 | 0% |

Source: Initial-phase data. Counts are of unique people. The same person can have multiple incidental findings reported. A person is counted only once for each incidental finding. Where a person had multiple LDCTs and multiple records for coronary calcification, for example, they were counted only once in the figure for coronary calcification. The sample matches the sample used in the OR analysis.

The Lung Cancer Screening Programme is one of few to use multivariable risk models to define eligibility. This approach was chosen based on evidence showing that risk models are more efficient at identifying participants with lung cancer. More recent research confirms this finding[19–21]. The lung cancer detection proportion of 1.4% seen in this program is lower than that observed in the UK Lung Screening trial (~2%) and UK pilots, and may reflect the wider coverage of the population. Furthermore, the proportion was calculated using the prevalent round CT number as the denominator, which may overestimate compared with the pilots. However, at this point, most lung cancers were detected at baseline or during surveillance CTs. Nevertheless, the cancer proportion exceeds that reported from the USA, even before eligibility criteria were expanded further in the USA[22]. The decision to use two rather than a single risk model was made in view of the lack of prospective head-to-head comparison between these two models in terms of cancer yield and cost-effectiveness in the UK population at the time the program commenced. Whole-program data showed that of those attending an LHC 47% were assessed as high risk and 90.5% of those underwent LDCT.

The participant-level analysis in the 'initial-phase' data (Extended Data Table 1 and Fig. 3) showed that slightly fewer females attended an LHC overall, and fewer were assessed as high risk. This merits further analysis to establish whether this is simply explained by fewer high-risk women responding to an LHC, as seen in some trials[23], or failure of risk prediction models to correctly predict risk. In addition, there was a slightly lower chance of undergoing LDCT in females who were assessed as high risk. LHC uptake was significantly lower in the more deprived, although these participants were more likely to be both at high risk and to have screen-detected cancer. The latter is expected and probably explained by the higher rates of smoking that would increase the risk estimates from both multivariable models. People from white ethnic backgrounds were markedly more likely to attend their LHC than people with other ethnicities (62.5% versus 18.8%). These data should be interpreted with caution because they represent participants in the earlier wave of invitations and consequently may differ from whole-program aggregated data where overall participation is higher. However, screening uptake is recognized to be affected by ethnicity in other screening programs[24], and the marked difference for ethnicity demands close monitoring and should inform elements of future LHC design addressing inequities[25,26]. Additional gains in uptake and cancer detection may then be possible by delivering engagement strategies targeted at these underserved groups. Projects have trialed a wide range of approaches, with mixed results, but there has been insufficient assessment of efficacy, making this a topic for future research.

The increase in early-stage proportion shown by national lung cancer registration data in line with screening activity and the finding that this was most marked in the most deprived socioeconomic quintile of the population shows what countries can expect from lung cancer screening. The change in socioeconomic distribution of early-stage disease may reduce as the program is rolled out to areas with lower incidence of lung cancer, but the targeted nature of the program and the use of multivariable models should mitigate this by identifying those at higher risk who will inevitably be more likely to come from the more deprived sectors of society.

Early stage detection is subject to overdiagnosis; what is needed to confirm efficacy, is a reduction in late-stage rate and mortality. Both of these take longer to become apparent. However, a report from Manchester, one of the earliest sites, confirmed a 25% difference in late stage between geographical areas with and without screening[27], which is correlated with mortality reduction[28].

An important challenge has been the management of non-lung-cancer diagnoses and other incidental findings. From the outset, this was supported by an incidental findings protocol as part of the quality assurance standard[4]. The principle underlying the protocol is to act only on those findings where there is likely to be benefit. For example, the most common finding, coronary artery calcification, is managed assuming that most people eligible for screening will also be eligible for primary prevention. Many patients have not had this; therefore, the findings of moderate and severe calcification prompt a reminder to the participant and primary care team. Currently no additional action is recommended for mild calcification because the evidence for benefit in the context of screening is not confirmed[29].

The program has quantified how common incidental findings are; while this provides opportunities to improve outcomes, management must also address overdiagnosis, physical and psychological harm, risk to program delivery and impact on workload in primary and secondary care. A study showed that despite a protocol being in place, many participants did not receive the recommended interventions[30]. In response, the latest version of the protocol includes nationally standardized pathways for the management of incidental findings. The incidental findings protocol is tailored to the UK healthcare system and, after legal opinion, addresses the medico-legal implications of the management of incidental findings. This was an important step in reassuring radiologists about the safety of following the protocol. It is an important country-specific aspect of management of incidental findings.

The screening program has put existing services under some pressure and required an increase in surgical and oncology capacity

to manage the additional curative-intent treatment. As the program expands further, new sites are able to learn from earlier adopters about the requirement for expansion of services and workforce. The program has used locally outsourced capacity for LHC call provision and LDCT scanning, including mobile unit provision and radiology reporting. Computer-aided detection is already mandated for lung nodules and often includes automated volumetry. A future challenge is to ensure that lung screening meets the standards set for screening in the NHS, in particular robust quality assurance of performance. This in turn requires improved data systems bespoke to lung screening.

The main limitation of the analyses presented is that data collection is 'real world' and may not be as rigorously verified as in a research trial. However, the very large dataset presented mitigates this as there are unlikely to be systematic errors; any individual inaccuracies are unlikely to affect the overall analysis. Detailed participant record-level data with cancer diagnoses linked to national registration data were only available for the initial phases of the program, so this means that findings on demographic impact on screening participation are less reliable. For participant-level data, we censored dates for LDCT follow-up at 185 days for diagnoses related to screening, recorded through linked national registry data accepting that this may result in some underrepresentation of scan totals and attributable cancer diagnoses.

Missing data on ethnicity and problems with completeness of smoking records in primary care are an important challenge for successful implementation of lung cancer screening; this is often a greater challenge in other countries. Although NHS primary care records have relatively high levels of smoking record accuracy, for some individuals direct invitation to establish smoking status is required[31].

Limited data were available for some important measures of performance. These include data on recall rate, incidental findings referral rate, details of nodule findings required to measure false positives, interval cancers and data on outcomes. This detail will be part of future publications, but some radiology reporting consortia collect these data and use this to feedback on individual radiology performance. NHS England has developed 14 effectiveness standards, currently in consultation, which are designed to provide a basis for performance management and quality assurance of outcomes.

The program does not currently have a bespoke, end-to-end participant-level information technology system, although this is recommended in the Standard Protocol[3]. This is expected in mid-2026.

The program presents an important opportunity for research and several studies have developed alongside the implementation, using datasets considerably larger than available from trials[25,32].

In conclusion, the NHS England national Lung Cancer Screening Programme has shown how large-scale implementation can be achieved at speed in high-risk groups through the application of a single protocol and effective project management, informed by previous research and smaller-scale pilots. Almost a third of the eligible population has been invited, with evidence of downstream impacts on the stage of diagnosis in participants and national lung cancer data. In doing so it demonstrates what can be expected and provides practical tools adaptable for use in other countries. The program has achieved these results despite other significant challenges in the NHS, including workforce and economic strain, industrial action and a respiratory pandemic. These data, and the material available freely online, and as supplementary material, provide evidence of feasibility, of the impact of a larger-scale program and a blueprint to assist others implementing this essential element of lung cancer care.

## Online content

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

[1]The Early Diagnosis and Detection Centre, The Royal Marsden NHS Foundation Trust, London, UK. [2]Institute of Cancer Research, London, UK. [3]National Heart and Lung Institute, Imperial College London, London, UK. [4]NHS England, London, UK. [5]University College London, London, UK. [6]Manchester University NHS Foundation Trust, Manchester, UK. [7]Division of Immunology, Immunity to Infection & Respiratory Medicine; Faculty of Biology, Medicine and Health, University of Manchester, Manchester, UK. [8]The Strategy Unit, NHS Midlands and Lancashire Commissioning Support Unit, Birmingham, UK. [9]Ipsos UK, London, UK. [10]Division of Population Medicine, School of Medicine, Cardiff University, Cardiff, UK. [11]Hull University Teaching Hospital, Hull, UK. [12]Royal Brompton and Harefield Hospitals, London, UK. [13]Roy Castle Lung Cancer Foundation, Liverpool, UK. [14]Patient Representative, Stockbridge, UK. [15]UCL Respiratory, University College London, London, UK. [16]University of Southampton, Southampton, UK. [17]UK National Screening Committee, London, UK. [18]Wolfson Institute of Population Health, Queen Mary University of London, London, UK. [19]Dorset County Hospital, Dorset, UK. [20]Patient Representative, Sandwell, UK. [21]Department of Oncology, University of Cambridge, Cambridge, UK. [22]Royal Papworth Hospital NHS Foundation Trust, Cambridge, UK. [23]Leeds Teaching Hospitals, Leeds, UK. [24]Leeds Institute of Health Sciences, University of Leeds, Leeds, UK. [25]Nottingham University Hospitals, Nottingham, UK. [60]These authors contributed equally: Richard W. Lee, Arjun Nair. *A list of authors and their affiliations appears at the end of the paper. ✉e-mail: Richard.Lee@rmh.nhs.uk

**UK Lung Cancer Screening Research Consortium**

Ellis Akhurst[9], Asif Azam[26], Thilan Rajith Bartholomeuz[27], Stuart Baugh[28], Anna Bibby[29], Richard Booton[6], Richard Brindle[4], Leanne Cheyne[30], Cyrus Daneshvar[31], Dhananjay Desai[32], Jennifer Lisa Dickson[33], Gurnak Singh Dosanjh[34], Jacqueline Faccenda[35], Elizabeth Ruth Fuller[36], Matt Gabriel Gallardo[37], Sindy Gill[38], Jennifer Graves[39], Seamus Grundy[40], Christopher Hale[9], Alexander Hicks[41], John Howells[42], Ian Hume[43], Martin Ledson[44], Kai En Low[45], Carolyn Mackinlay[46], Afsar Madathil[47], Venkateswaran Mahadevan[47], Nicola McMaster[4], Stephen McSwiney[9], Jenny Messenger[37], Carol Min[48], Peer Mohamad Mohamed[49], Babu Naidu[50], Kofi Nimako[51], Jason Page[52], Jason Poole[4], Helen Powell[38], Baiju Saradananda Prasad[53], Arvind Rajasekaran[54], Poppy Richards[4], Rabinder Singh Randhawa[53], Kimuli Ryanna[55], Charlotte Smith[4], Haarini Sridhar[4], Rebecca Taylor[56], Stephanie Uys[57], Anna C. Walsham[40], Christopher James Warburton[58], Ann Ward[36], Mark Weatherhead[59] & Tim Windle[4]

[26]The Dudley Group NHS Trust, Dudley, UK. [27]Nottingham and Nottinghamshire Integrated Care Board, Nottingham, UK. [28]Northern Lincolnshire and Goole NHS Foundation Trust, Scunthorpe, UK. [29]University of Bristol, Bristol, UK. [30]Bradford Teaching Hospitals NHS Foundation Trust, Bradford, UK. [31]University Hospitals Plymouth NHS Trust, Plymouth, UK. [32]University Hospital Coventry and Warwickshire, Coventry, UK. [33]West Suffolk NHS Foundation Trust, Bury St Edmunds, UK. [34]Leicestershire and Rutland Integrated Care Board, Leicester, UK. [35]North West Anglia Foundation Trust, Peterborough, UK. [36]The Newcastle upon Tyne Hospitals NHS Foundation Trust, Newcastle, UK. [37]University Hospitals Sussex NHS Foundation Trust, Brighton, UK. [38]Frimley Health NHS Foundation Trust, Camberley, UK. [39]Dorset County Hospital NHS Foundation Trust, Dorset, UK. [40]Northern Care Alliance NHS Foundation Trust, Salford, UK. [41]Portsmouth Hospitals University NHS Trust, Portsmouth, UK. [42]Lancashire Teaching Hospitals, Lancashire, UK. [43]Norfolk and Waveney Integrated Care Board, Norfolk, UK. [44]Liverpool Heart and Chest Hospital, Liverpool, UK. [45]Bedfordshire Hospitals NHS Foundation Trust, Bedfordshire, UK. [46]Great Western hospitals NHS Foundation Trust, Swindon, UK. [47]James Paget University Hospital NHS Foundation Trust, Norfolk, UK. [48]George Eliot Hospital, Nuneaton, UK. [49]East and North Hertfordshire NHS Trust, Stevenage, UK. [50]University of Birmingham, Birmingham, UK. [51]Surrey

and Sussex Healthcare NHS Trust, Redhill, UK. [52]South Yorkshire Integrated Care Board, Sheffield, UK. [53]Milton Keynes University Hospital, Milton Keynes, UK. [54]Sandwell and West Birmingham NHS trust, Birmingham, UK. [55]Guy's and St Thomas' NHS Foundation Trust, London, UK. [56]South Tyneside and Sunderland, NHS Foundation Trust, Sunderland, UK. [57]St Bartholomew's Hospital, London, UK. [58]Cheshire & Merseyside Cancer Alliance, Liverpool, UK. [59]Northumbria Healthcare NHS Trust, Newcastle, UK.

## Methods

### Approvals and ethics statement

This work complies with all relevant research and information governance and ethical regulations. The work involved the analysis of data from the TLHC Programme and routine wider NHS, collected as part of a defined service evaluation within NHS England. In accordance with the UK Health Research Authority 'Decision Tool' and associated guidance on differentiating research from service evaluation, this project was classified as service evaluation; therefore, it did not require review by an NHS Research Ethics Committee. The primary purpose of the project was to assess and improve existing service delivery, without randomization.

Approval confirming that this work falls under service evaluation was obtained through NHS England's internal governance processes, in consultation with the TLHC Expert Advisory Group, and the Cancer Programme's Evaluation Oversight Group. All data used were pseudonymized before analysis; no identifiable data were accessed at any stage. All analyses were performed under NHS England's statutory responsibilities for service evaluation, data protection and information governance. Individual-level patient consent was not required. The legal basis for patient-level data linkage was obtained by NHS England Cancer Programme Pilots Evaluation Directions 2024[33]. Directions were given by the Secretary of State for Health and Social Care to require NHS England to establish an information system for the collection and analysis of data to evaluate the effectiveness of the NHS England Cancer Programme Pilots.

### Screening methodology

All sites followed a standardized protocol and quality assurance standards, except for research and pilot studies that were already running and retroactively incorporated into the program. The 'Standard Protocol prepared for the Lung Cancer Screening Programme' and 'Quality Assurance standards prepared for the Lung Cancer Screening Programme' were revised in 2025, but without substantial change to the program[3,4]. Briefly, people aged 55–74 at invitation, and registered at a primary care practice with electronic or self-reported, current or former smoking history, are potentially eligible. These potential participants are contacted by letter, SMS or telephone and invited to an LHC to assess their risk of developing lung cancer. LHCs are either undertaken by phone, with subsequent face-to-face assessment in those with a risk of lung cancer above the defined threshold, or face to face. The LHC also includes a screen for respiratory and cancer symptoms, basic medical history, recent CT scan imaging (<12 months) and tobacco dependency assessment and treatment. Some LHCs include spirometry and cardiovascular risk assessment. Most of the LHCs are performed in community-based settings, typically using bespoke mobile units. Others are performed in primary or secondary care establishments. Full inclusion and exclusion criteria are detailed in the Standard Protocol[3]. Exclusion criteria include factors that would contraindicate curative treatment, such as frailty and metastatic cancer.

The lung cancer risk assessment is based on two externally validated multivariable models, that is, the $PLCO_{m2012}$ models and the $LLP_{v2}$ (refs. 5,19). LHC participants are eligible for LDCT screening if their risk of lung cancer meets or exceeds predefined thresholds for either model ($PLCO_{m2012} \geq 1.51\%$ over 6 years or $LLP_{v2} \geq 2.5\%$ over 5 years). The LDCT is offered either in community mobile scanning units or on fixed site scanners. All screening LDCT scans are reported by thoracic radiologists with lung cancer expertise who regularly participate in lung cancer multidisciplinary team meetings, with centralized training in lung cancer radiology screening principles. Radiologists are subject to quality assurance through a bespoke external quality assurance system, the PERFormance Evaluation for CT Screening program in line with international consensus recommendations[34,35]. Volumetry and computer-aided detection of lung nodules are mandated. British Thoracic Society guidelines are followed for lung nodule management[36], modified to include guidance on new nodules discovered on interval or incident round screening[37]. Lung screening pulmonary nodule guidance classifies all lung nodule findings into one of three categories: negative—return to next screening round; positive—refer for investigation; or indeterminate—interval surveillance at 3 and 12 months from baseline, within the Lung Cancer Screening Programme (additional CT at 24 months from baseline if volumetry not possible). Participants with positive findings (suspicious for lung cancer) are peer-reviewed at multidisciplinary 'Screening Review Meetings' before referral to participating local lung cancer services where management follows the National Optimal Lung Cancer Pathway[38]. Participants with LDCT scans having no actionable findings are returned to the next screening round. Incidental findings are managed according to guidance in the Quality Assurance Standard appendix[4]. Ongoing screening is biennial if there are no new actionable findings, until participants age out of screening. The stratification to biennial screening for participants with no actionable nodules was based on the findings in both the National Lung Screening Trial and NELSON where the cancer detection rate was twofold to threefold less at the next annual screen than when nodules were present[39,40]. Although biennial strategies are not on the cost-effectiveness frontier, they are both more affordable and more deliverable given capacity constraints[12]; the strategy adopted was judged to allow a greater proportion of the population to be screened.

### Site selection

The aim of the program was to evaluate and sustain implementation of lung screening. Initial sites were selected using data on lung cancer for individual healthcare commissioners (then termed Clinical Commissioning Groups). Geographical areas with the highest lung cancer incidence or mortality were selected with the intention of focusing the start of the lung cancer screening (then TLHC) program in areas with the greatest need. Local healthcare systems were supported in setting up local and national governance, data sharing agreements, local demand modeling, choice of implementation model (for example, virtual or face to face, or in-house versus outsourced), pathways and protocols for downstream referrals and incidental findings, recruitment advice, staff training and developing local strategies for rollout. Funding was contingent on LHC/LDCT scan delivery volumes. Local clinical and operational factors and budgetary arrangements determined specific local delivery models within the constraints of the program's protocol and governance. The initial phase consisted of 14 sites with first invitations in July 2019, which was thereafter expanded to all regions in England. The planned rollout was halted by the COVID-19 pandemic but resumed with adaptations for infection control by using telephone triage, hygiene measures and stopping spirometry.

### Data

Data are presented in two ways to capture the program scale but reflecting greater detail in 'initial-phase data', collected by 13 (of 14) of the initial-phase sites where individual participant record-level detail was available as part of the national evaluation. Whole-program data are simpler to collect, so achievable by all clinical sites, summarizing data in aggregate, nationally. More focused data, including a 'patient-level identifier' (NHS number) were required from the 'initial-phase sites', which were resourced to provide these deeper data. Only in patients with an NHS number, was it possible to perform linkage to other data. This hybrid approach was taken because of the scale of this program and data resource available. The initial-phase data are also presented restricted to those participants who were followed up for a period of 185 days. This was the time after LDCT when any lung cancer diagnosed could be reliably attributed to the LDCT (Table 1 and Extended Data Fig. 1). The initial-phase data represent a subset of 'whole-program data', as below. No statistical method was used to predetermine sample size.

**Whole-program data.** Whole-program data were collected from all eligible participants according to predefined variables determined for each component of the clinical pathway, designed to measure the end-to-end screening process and outcomes. Data were obtained from program start to March 2025 and analyzed by NHS England. Trajectory data are derived from the regional 'Cancer Alliance', arm's length strategic alliances that fund cancer innovation, who provided planned or modeled annual activity. The eligible population is a modeled figure expected in 2028–2029, including mitigations for caveats such as people aging in and aging out. Cancer detection is expressed as a proportion of the prevalence around LDCTs (equivalent to 'per participant'). All cancers in the prevalence and incident rounds are included. Incidental findings were reported according to predefined codes determined by NHS England.

**Initial-phase data.** A detailed participant-level minimum dataset (MDS) was collected from the initial project sites (Extended Data Fig. 1) as part of service evaluation. Data were collected from all eligible participants according to predefined variables determined for each component of the clinical pathway, designed to measure the end-to-end screening process and outcomes. These returns were used to monitor the progress of the TLHC pilot program. The record-level datasets were linked to cancer diagnosis and staging data from NCRAS, including COSD. Specifically, a combination of the 'gold standard' cancer registry, also known as the NCRD, and Rapid Cancer Registration (RCRD) datasets were used[41]. The datasets are linked to the NHS number; then, logic is applied to assign the correct cancer diagnosis to the correct activity data. The patient-level analysis uses the MDS to track participant pathways from invite to CT scan. The MDS is then linked to the combined NCRAS dataset to show participant outcomes (diagnosis and staging rates). This analysis is based on TLHC activity data submitted up to March 2024 and NCRAS lung cancer activity to August 2023. Rules were applied to remove any double counting and manage data quality issues. The process for linking these activities can be provided on request.

Of note, we included a 185-day limit as a classification method. Diagnoses considered attributable to lung screening were those that followed a CT scan or an LHC resulting in a high-risk assessment and were diagnosed within 185 days of the above event. This represents the maximum acceptable time between a person's lung cancer screening scan and subsequent diagnosis to consider the two events as directly linked, that is, we attributed a lung cancer diagnosis to the screening process if it occurred within 185 days of the scan. Our method for establishing this limit involved survival analysis to assess the likelihood of diagnosis following a scan and time-to-event segmentation. This process was quality-assured by a member of the NCRAS team.

Censoring in this analysis results from two factors: the 185-day classification limit and the cancer registry data ending before the lung cancer screening activity data. Specifically, the cancer registry data ended in August 2023, which was earlier than the 'initial-phase' data submissions ending in March 2024. This created a scenario where some participants were scanned between September 2023 and March 2024 but we could not determine their lung cancer diagnosis status. August 2023 became our 'hard-censor' point as no diagnoses were recorded after this date. The 185-day classification introduced additional complexity. Participants scanned up to 184 days before the August cutoff without a cancer registry record could not be classified as either lung-cancer-free or having a lung-cancer-screening-attributable diagnosis. Some participants scanned during this period had a lung cancer diagnosis made within the 185-day limit and were counted as attributable to lung screening. A 'soft-censor' period ran up to August 2023; it covers people scanned and diagnosed within 185 days leading up to August 2023.

The results of this more detailed initial phase include the prevalence round and an incidence round up to 24 months from baseline. Data are presented for the combined outcome of all rounds and follow-up. Non-lung-cancers included in the analysis are attributed to the program based on linking LDCT data to cancer registry data. The length of time from scan to diagnosis and cancer type are included to allow readers to make a judgment on whether this arises from lung cancer screening or through an alternative clinical pathway.

## Program impact on lung cancer in England
RCRD on lung cancer in England was used to look at the incidence and early stage detection. This dashboard enables analyses according to tumor site and socioeconomic quintile. In this analysis, data were used up to March 2025. RCRD provides a quicker indicative source of cancer data compared to the NCRD, which is the 'gold standard' registration dataset and which relies on additional data sources, enhanced follow-up with trusts and expert processing by cancer registration officers.

An evaluation partner (Ipsos UK) and a Commissioning Support (The Strategy Unit) were commissioned to oversee data collection and conduct a service evaluation. Data covering the whole-program activity was collected by the North of England Care System Support Unit via monthly aggregate data returns. Data for the service evaluation was collected from the initial project sites by the Midlands and Lancashire Care Commissioning Support Unit via monthly patient-level data returns. Additional quarterly reports covering the quality assurance standards were collected by NHS England.

## Statistics and reproducibility
**Statistical analysis.** Two main analyses were undertaken. First, whole-program descriptive data were analyzed, including demographics, screening uptake (proportion attending an LHC of those eligible for invitation), eligibility for LDCT according to $LLP_{v2}$/$PLCO_{m2012}$ risk score, LDCT surveillance rates and screen-detected lung cancer data. Second, initial-phase data were used to assess the associations between demographic characteristics (age, sex, ethnicity and Indices of Multiple Deprivation quintile) with the likelihood of people in the eligible population attending an LHC, receiving an LDCT and having a lung cancer diagnosis.

Further analysis was conducted using binary logistic regression, having assessed for separation and collinearity (no issues were found). Four multivariable logistic regression models were conducted on separate samples: (1) LHC attendance of those who were eligible for invitation based on primary care records; (2) LHC outcome high risk of those who attended an LHC; (3) LDCT attendance of those who attended an LHC and were high risk; and (4) lung cancer detected of those who attended an LDCT scan at least 185 days before the end of the COSD cancer data. All demographic variables were included in each regression model; because of the high number of missing demographic data recorded in the pilot sample, separate categories were retained for age, ethnicity or deprivation not known. Because of differences in missing data in people who attended an LHC and an LDCT, data selected for regression analysis (Fig. 3) differ slightly to that in Extended Data Table 1. This is because records with complete data for both the starting point (for example, attendance at an LHC) and the end point (for example, undergoing LDCT) form a slightly different subset than those with complete data for each variable considered independently.

All available cases were included from the start of the program. Curation of each dataset is described above.

## Software
R v.4.4.0 onwards, and binomial logistic regression modeling using a generalized linear model from the stats package, were used.

## Reporting summary
Further information on research design is available in the Nature Portfolio Reporting Summary linked to this article.

## Data availability

The aggregate whole-program data (April 2019–March 2025) and eligibility population data used in Figs. 1 and 2a are not publicly available. These data are managed and stored by NHS England for the purpose of monitoring lung cancer screening activity. Aggregate data may be made available upon request to the NHS England Lung Cancer Screening Programme team (lungcancer.screening@nhs. net). The RCRD was used to make Fig. 2b. This is available in the public domain. RCRD dashboards – National Disease Registration Service. The record-level initial-phase data (April 2019–March 2024) used in Fig. 3, Extended Data Table 1 and Table 2a,b contain confidential patient-level information and cannot be shared publicly because of the data protection and information governance requirements set up for the pilot. These data are held securely by NHS England and access is restricted to safeguard patient privacy. All data access requests will be considered in accordance with NHS England data governance policies. Requests will be reviewed and responded to within 4–6 weeks.

## Code availability

The analysis code for this evaluation, from data submission to processed tables, is publicly available via GitHub at https://github.com/craig-parylo/610_tlhc.

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

## Acknowledgements

We thank the extensive efforts by local site clinical and operational leads, the Midlands and Lancashire Care Commissioning Support Unit, Ipsos UK, NHS England analysts and cancer program team, expert advisory group and wider NHS England Leadership and cancer program team, and patient and public representation, which aided program funding, design, implementation and delivery, oversight, and data curation and analysis. This paper represents independent research supported by (1) the Royal Marsden Cancer Charity and (2) the National Institute for Health and Care Research (NIHR) Biomedical Research Centre at the Royal Marsden NHS Foundation Trust and The Institute of Cancer Research, London. The funders had no role in study design, data collection and analysis, decision to publish or preparation of the manuscript. The authors received no specific funding for this work.

## Author contributions

R.W.L., A.N., D.R.B., H.B. and C.G. conceived the study and developed the main conceptual ideas. R.W.L., A.N., D.R.B., M.C. and H.B. developed the proof outline. C.G., C.P., J.A. and Ipsos performed the data acquisition and preparation. C.G., C.P., J.A. and R.W.L. undertook the analysis. C.G., J.A. and C.P. vouch for data quality and reliability. R.W.L. and D.R.B. reviewed the methodology. R.W.L., A.N., D.R.B., M.C. and H.B. wrote the paper. R.W.L., A.N., C.G., J.A., D.R.B., M.C., A.D., H.B., N.N., R.C.R., P.C., J.F., S.M.J., S.L.Q., K.B., M.W., M.L., S.M., M.G., P.J., A.M., A.R., L.R., P.S., M.E.J.C., and J.R. edited and supported the methodology and analysis. The individuals listed in the UK Lung Cancer Screening Research Consortium contributed to program delivery, data curation, development of the methodology and paper direction.

## Competing interests

H.B. reports honoraria or institutional funding for nonpromotional educational talks or advisory boards from Intuitive Surgical, AMBU, Verathon, ESMO, NICE and OncLive, as well as paid proctorship work with Intuitive Surgical. M.G. has served as a member of the NHS England Screening Advisory Group, the chair of trustees for the British Thoracic Oncology Group and UK Lung Cancer Coalition as a trustee for the Roy Castle Lung Cancer Foundation. R.W.L. is funded by the Royal Marsden NIHR Biomedical Research Centre (BRC) and the Royal Marsden Cancer Charity. R.W.L.'s institution receives compensation for time spent in a secondment role for the NHS England Lung Health Check program and previously as National Specialty Lead for the NIHR. He has received research funding from Cancer Research UK (CRUK), Innovate UK (cofunded by GE HealthCare, Roche Diagnostics, Optellum, Elliptica and RNA Guardian) and SBRI Healthcare (including as a co-applicant with QURE.AI), RM Partners Cancer Alliance and NIHR (including co-applicant in grants with Optellum). He has received honoraria, speaker/advisory fees, and hospitality/ travel expenses from CRUK, Roche Diagnostics, Johnson & Johnson, Guardant, AstraZeneca and King Faisal Hospital, Saudi Arabia. He also undertakes medical private practice. S.L.Q. is supported by Barts Charity (G-001522, MGU0461) and declares grant income from the NIHR, National Institutes of Health, Yorkshire Cancer Research and CRUK. R.C.R. is funded by Cambridge NIHR BRC (NIHR203312) and CRUK (Cambridge Centre CTRQQR-2021/100012). He receives research funding from CRUK, UKRI Medical Research Council (MRC), and Asthma and Lung UK. He is a director of and clinical lead for the UK Lung Cancer Coalition. He is a clinical director for the Cambridge and Peterborough Integrated Care System (ICS) Lung Cancer Screening Programme. N.N. is supported by an MRC Clinical Academic Research Partnership (MR/T02481X/1). This work was partly undertaken at University College London Hospitals and University College London, which received a proportion of funding from the Department of Health's NIHR BRC funding scheme. He has received research funding from the NIHR, CRUK, SBRI, UKRI and the Ruth Strauss Foundation. N.N. reports honoraria or institutional funding for nonpromotional educational talks or advisory boards from Amgen, AstraZeneca, AXANA, Boehringer Ingelheim, Bristol Myers Squibb, EQRx, Fujifilm, Guardant Health, Intuitive, Janssen, Lilly, Merck Sharp & Dohme, Olympus, Roche and Sanofi. S.M.J. has received grant income from GRAIL Inc. He is an unpaid member of a GRAIL advisory board. He has received lecture fees for academic meetings from AstraZeneca. His wife works for AstraZeneca. The SUMMIT study is funded by GRAIL through a research grant awarded to S.M.J. as principal investigator. S.M.J. is supported by CRUK program grant no. EDDCPGM/100002 and an MRC program grant (MR/W025051/1/MRC), the Rosetrees

Trust, the Roy Castle Lung Cancer foundation, the Garfield Weston Trust and the UCLH Charitable Foundation. A.R. declares consulting fees from Roche. J.R. was a member of the NHS England screening advisory group and is a member of the UK LC clinical expert group, British Thoracic Oncology Group steering committee, the ELF/ERS LC task forces and screening groups, the European Organisation for Research and Treatment of Cancer LC group and patient panel, the Cancer research advocates forum, the UCL lung cancer patient and public involvement (PPI) group, the Oxford cancer PPI group, the WM Cancer Alliance LC EAG and advocates' EAG and BSOL ICS Targeted Lung Health Check working group. She is involved in PPI in several LC studies including PERFormance Evaluation for CT Screening (Nottingham), Lung Impact, DART and DOLCE, which use several industry software programs and tools for lung imaging, including Optellum and QURE AI. She has received occasional honoraria/travel expenses for LC screening meetings in the UK and Europe. D.R.B. declares grant income from Innovate UK, CRUK, SBRI and Horizon, and lecture honoraria from Boehringer Ingelheim and AstraZeneca. P.C. declares grant income from the NIHR, CRUK, Innovate UK, Yorkshire Cancer Research and Greater Manchester Cancer Alliance, consulting fees from Roche and honoraria from Bayer and Novartis. A.D. reports consulting fees from Boehringer Ingelheim, Roche, AstraZeneca and stocks interests in Brainomix. The other authors declare no competing interests.

## Additional information

**Extended data** is available for this paper at https://doi.org/10.1038/s41591-026-04292-y.

**Correspondence and requests for materials** should be addressed to Richard W. Lee.

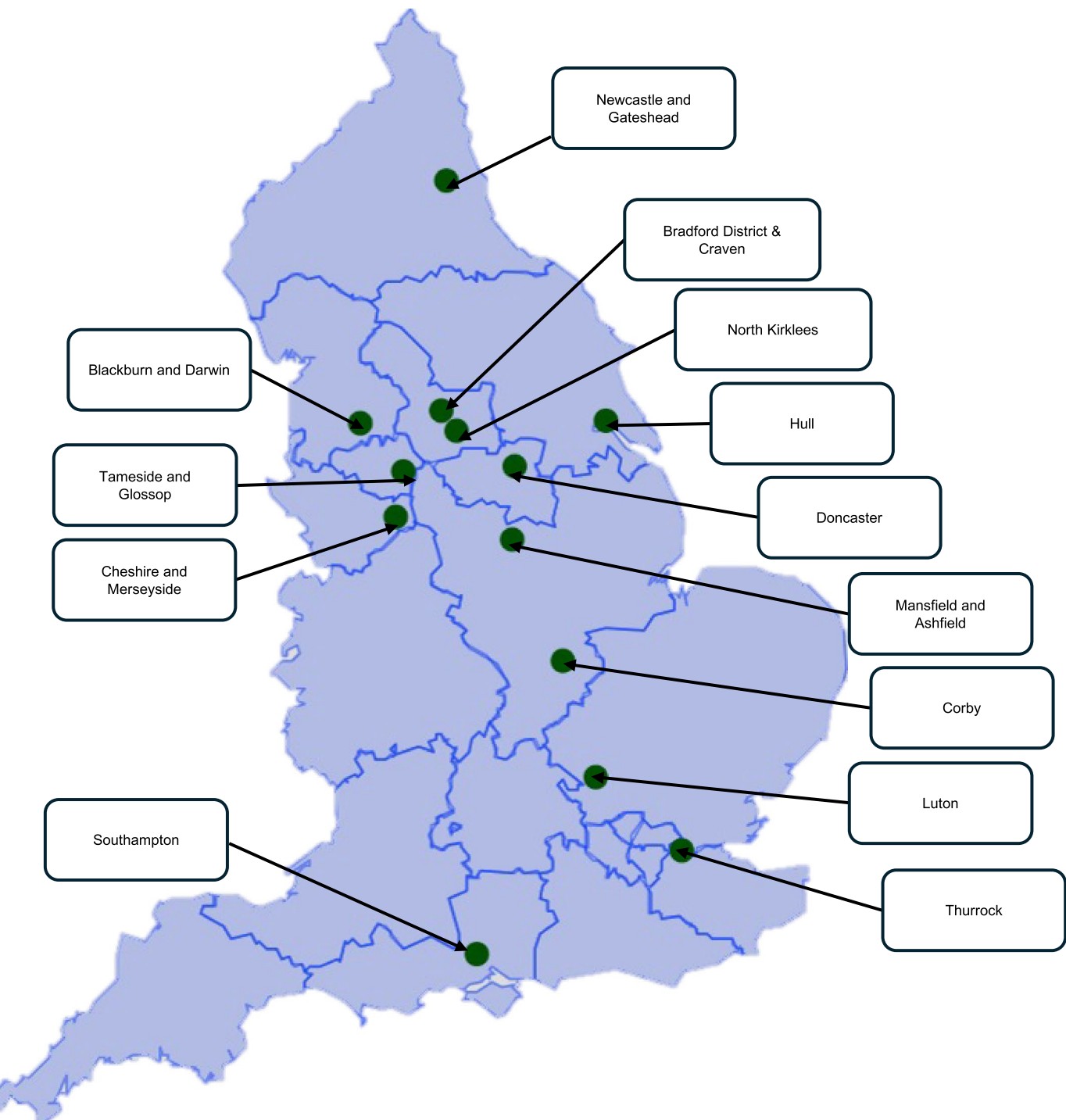

**Extended Data Fig. 1 | Initial Project Site Locations.** Map of England indicating Initial Project Site locations: 'Initial-Phase' Participant record-level data are available from these sites.

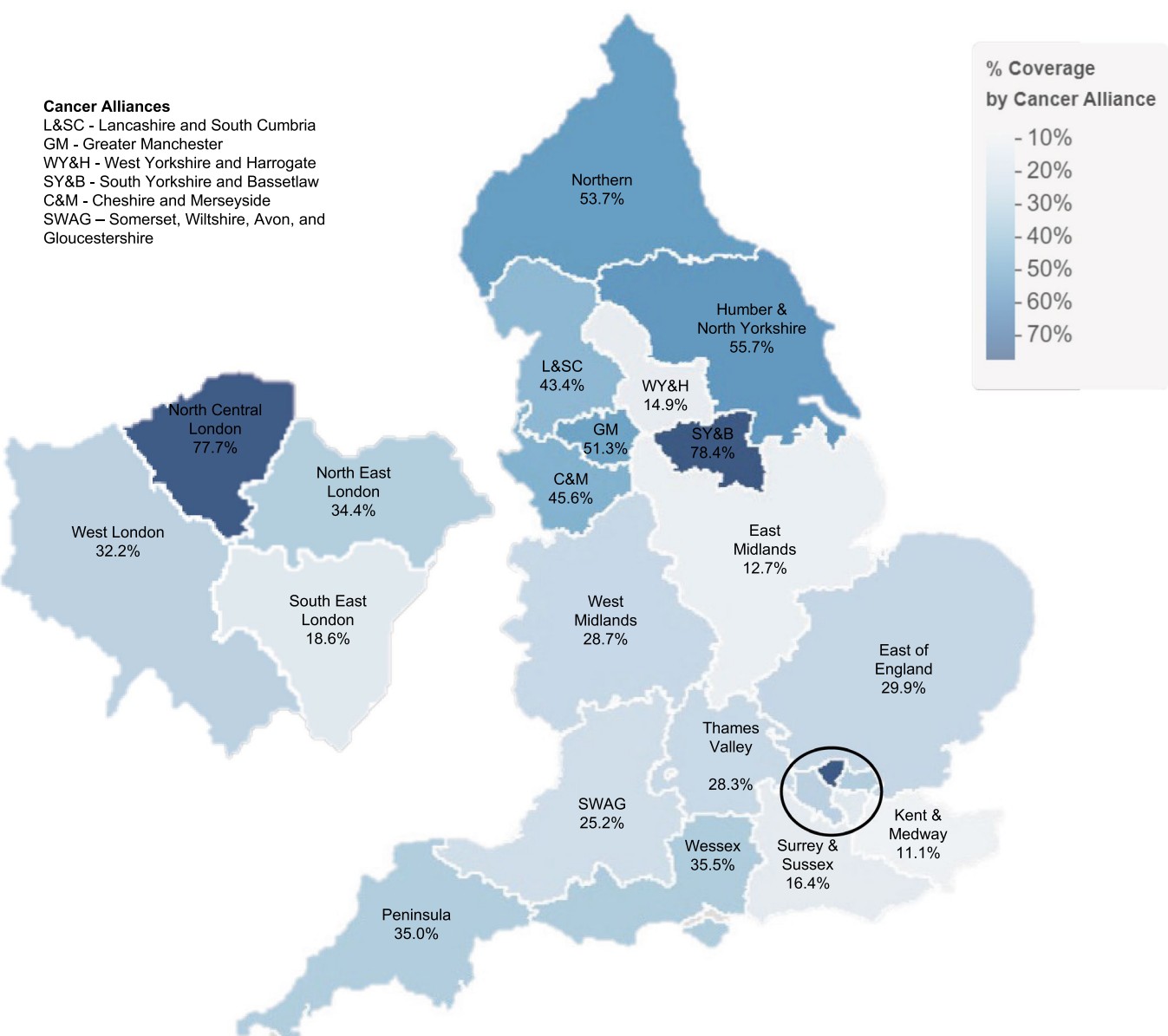

**Cancer Alliances**
L&SC - Lancashire and South Cumbria
GM - Greater Manchester
WY&H - West Yorkshire and Harrogate
SY&B - South Yorkshire and Bassetlaw
C&M - Cheshire and Merseyside
SWAG – Somerset, Wiltshire, Avon, and Gloucestershire

% Coverage by Cancer Alliance
- 10%
- 20%
- 30%
- 40%
- 50%
- 60%
- 70%

Northern 53.7%

Humber & North Yorkshire 55.7%

L&SC 43.4%

WY&H 14.9%

North Central London 77.7%

North East London 34.4%

West London 32.2%

GM 51.3%

SY&B 78.4%

C&M 45.6%

East Midlands 12.7%

South East London 18.6%

West Midlands 28.7%

East of England 29.9%

Thames Valley 28.3%

SWAG 25.2%

Wessex 35.5%

Surrey & Sussex 16.4%

Kent & Medway 11.1%

Peninsula 35.0%

**Extended Data Fig. 2 | NHS England Lung Cancer Screening Programme National Coverage.** Map indicating percentage national coverage by March 2025 (N invites sent/N eligible population 2028/29) by cancer alliance.

**Extended Data Table 1 | Initial Phase Odds Ratio Analysis Cohort Characteristics**

| Demographic group [a] | Eligible population | Attended LHC | % LHC/ Eligible | Attended LDCT | % LDCT/ LHC | Attended LDCT [b] (185 day f/up) | Lung cancers [c] (185 day f/up) |
|---|---|---|---|---|---|---|---|
| Total | 582,700 | 216,985 | 37.2% | 114,430 | 52.7% | 53,435 | 1,565 |
| **Age group** | | | | | | | |
| 55-64 | 294,500 | 96,695 | 32.8% | 41,880 | 43.3% | 18,175 | 330 |
| 65-74 | 240,395 | 91,385 | 38.0% | 59,985 | 65.6% | 26,670 | 860 |
| 75 | 10,495 | 3,615 | 34.4% | 3,040 | 84.1% | 1,465 | 40 |
| Other | 8,380 | 3,490 | 41.6% | 2,945 | 84.4% | 1,550 | 60 |
| Not known | 28,930 | 21,800 | 75.4% | 6,580 | 30.2% | 5,570 | 275 |
| **Sex** | | | | | | | |
| Male | 303,825 | 113,670 | 37.4% | 64,440 | 56.7% | 29,995 | 730 |
| Female | 275,335 | 103,310 | 37.5% | 49,985 | 48.4% | 23,415 | 835 |
| Not known | 3,540 | <10 | - | <10 | - | 25 | <10 |
| **Ethnic group** | | | | | | | |
| White | 295,450 | 184,765 | 62.5% | 95,060 | 51.5% | 42,790 | 1,450 |
| Other ethnicities | 97,265 | 18,295 | 18.8% | 9,320 | 51.0% | 4,490 | 75 |
| Not known | 189,985 | 13,925 | 7.3% | 10,045 | 72.2% | 6,155 | 35 |
| **Deprivation** | | | | | | | |
| Quintile 1 | 258,500 | 85,285 | 33.0% | 48,760 | 57.2% | 23,475 | 820 |
| Quintile 2 | 118,490 | 44,610 | 37.7% | 22,565 | 50.6% | 10,260 | 330 |
| Quintile 3 | 79,725 | 31,515 | 39.5% | 15,110 | 47.9% | 6,690 | 210 |
| Quintile 4 | 68,620 | 28,550 | 41.6% | 12,495 | 43.8% | 5,360 | 145 |
| Quintile 5 | 36,385 | 15,050 | 41.4% | 5,665 | 37.6% | 2,010 | 60 |
| Not known | 20,985 | 11,970 | 57.1% | 9,835 | 82.2% | 5,640 | <10 |

Initial Phase Odds Ratio Analysis Cohort Characteristics of eligible population from which OR analysis carried out for 1) Lung Health Check (LHC attendance), 2) Low-Dose CT (LDCT) attendance and 3) lung cancers detected. f/up (follow up days observed for event). [a]Numbers are rounded to the nearest 5, counts below 10 are suppressed. [b]had a low dose CT thorax scan at least 185 days prior to the end of the COSD cancer data. [c]had a COSD linked lung cancer diagnosis (Lung screening-associated)

# Clinical data

All manuscripts should comply with the ICMJE guidelines for publication of clinical research and a completed CONSORT checklist must be included with all submissions.

| | |
|---|---|
| Clinical trial registration | N/A.  Real world data analysis as part of service evaluation and national registry data. https://www.england.nhs.uk/contact-us/privacy-notice/how-we-use-your-information/our-services/evaluation-of-the-targeted-lung-health-check-programme/ |
| Study protocol | https://www.england.nhs.uk/publication/targeted-screening-for-lung-cancer/ |
| Data collection | Monthly data submission to Data Service for Commissioners Regional Office (DSCRO) by providers of lung screening |
| Outcomes | Uptake of lung screening, uptake of CT scan, lung cancer diagnoses |

