## [Peer Review File · Nature Medicine]

Implementation of the NHS England Lung Cancer Screening Programme over 5 years

Corresponding Author: Dr Richard Lee

Version 0:

Reviewer comments:

Reviewer #1

(Remarks to the Author)

The authors provided a comprehensive and detailed account of the UK's Implementation of a National Lung Cancer Screening Programme (NLCSP) over the last five years. The success is seen in the actual numbers; over 2.5M individuals invited, 49% of whom attended an appointment (face to face or by telephone); over 500K have had their first LDCT scan and 7,193 lung cancers (1.4% cancer detection rate) have been identified by March 2025.

Thus, the NLCSP is an excellent example of successful implementation of lung cancer LDCT screening in the real world and is a credit to all the individuals involved, but particularly the individuals at the coal-face who have delivered the service on a daily basis, listed under the 'Lung Cancer Screening Programme Research Consortium'.

This article should not, however, solely be a report of a success story, as many of the international readers of this article will be setting up implementation programmes in their own countries or in the midst of feasibility studies.

Disappointingly, the authors have missed an opportunity to provide this guidance, which would significantly increase the value of this manuscript and make it more informative for an international readership.

The following points need further elaboration within the manuscript.

- Starting at the beginning – what evidence was utilised to even start the TLHC? The UKLS trial was the first RCT to utilise a lung cancer risk model to select high risk participants, this was followed by five local implementation projects (referenced in the manuscript); all of these implementation projects used either one or both of the two validated risk models (LLPv2 and or PLCO m2012). The UK National Screening committee (UKNSC) commissioned the cost effectiveness analysis of lung cancer screening, and it was only because of the use of risk models to recruit high risk individuals, that lung cancer screening was shown to be cost effective, <https://view-health-screening-recommendations.service.gov.uk/lung-cancer/>
This publication provides details of the analysis of the UK cost-effectiveness modelling, which resulted in the UKNSC recommending LDCT screening in targeted high-risk populations in 2022 and the national lung cancer screening was supported by the UK Prime Minister in 2023.

Unless this pivotal initial step is addressed as outlined above, those involved in the development of screening programmes will not have this crucial data as a basis for debating with their Ministry of Health, the necessary focus on targeted risk groups which are cost effective.

- The Stage 1 & II data provided in this manuscript is in the same ballpark as the UKLS trial and the five UK implementation projects, which is the best parameter of success to date (Fig 5), until the NHS LCS mortality data is available.
- There needs to be an explanation as to why the NHS LCS programme is run on a 24 month repeat basis, instead of 12 months, as the majority of the international RCTs were undertaken on a 12 month repeat basis. Only the NELSON trial included a two -year interval in their trial.

- The authors mention the issues around Incidental Findings (IF) and the impact on Primary care. The potential of utilising coronary artery calcification (CAC) as well as Emphysema / COPD data, is mentioned but needs much greater clarity in respect of what is proposed in the future (noted current status in Annex 2 in Ref 9). Internationally, IF is considered a major issue particularly in respect of how it will impact on their primary care providers. How will this be funded in the UK national screening programme and is it cost effective?
- Lung cancer screening at a national level, impacts on all of the workforce in the Lung Cancer screening pathway, however, the major impact is on the Radiology workforce. Are the current arrangements sustainable in the UK (some description should be included on the NHS/ private provision of radiology reporting), as well as the current use and future potential of incorporating artificial intelligence (AI) platforms in the reporting pathway. The NHS LSC programme is dependent on using mobile CT units, is this sustainable moving into a long-term national programme ?
- The discussion point: ' better data systems bespoke to the programme' .

The UK Biobank has provided international leadership in recruitment, collection of questionnaire data, clinical findings, together with data on a wide range of biomarkers; providing an invaluable resource for future clinical research. On reflection, should the TLHC/ NHS LCS programme not have considered utilising an integrated database with a central hub for the CT images, instead of the disparate system throughout the country? Can this situation be retrieved to improve access to future data and the CT images by the research community?

- Finally, the NHS LCS programme has succeeded in recruiting ~ 50% of the invited population, which is lower than other cancer screening programmes in the UK (acknowledge it's just starting) . However, the 50% which do not participate are mainly within the lower socio-economic deprived communities, who have a higher incidence of lung cancer, potentially as high as 2.5%. Surely this has to be considered and discussed in this manuscript?

Overall, this manuscript heralds the success story of Lung Cancer Screening in the UK but fails to provide a blueprint for screening in other countries and fails to address future challenges.

Other comments

References incomplete:

9. England. Standard protocol and quality assurance standards for the Lung Cancer Screening Programme. 2025.

15. NHS England. Implementing a timed lung cancer diagnostic pathway. 2024.

Does Figure S3 actually add anything to the manuscript?

(Remarks on code availability)

Reviewed the information on https://github.com/craig-parylo/610_tlhc

The information provided would require an individual with the necessary computer /IT experience in order to answer the question - are the results reproducible.

Recommend that you have a Reviewer with these IT skills to answer this question.

Reviewer #2

(Remarks to the Author)

Summary of the key results

The authors make an exceptional contribution to scientific developments in the field of (the implementation of) lung cancer screening by describing the first results of the national lung cancer screening programme as implemented in the UK. Although lung cancer screening is proven effective in reducing lung cancer/all-cause mortality, many challenges remain in the implementation of nationwide programmes.

For this programme, more than 2 million individuals were invited for a lung health check, which is according to the authors almost one third of the estimated eligible population for lung cancer screening in the UK. Over 7,000 lung cancers have been diagnosed, with a cancer detection rate of 1.4% of participants undergoing baseline scans. A favorable stage distribution towards early stages could be confirmed and are comparable with results of other lung cancer screening trials.

This paper also highlights other important aspects of lung cancer screening programme, such as the risk-based recruitment and screening uptake, with special attention to people living in the more deprived areas, where lung cancer incidence is relatively higher and where the impact of lung cancer screening might be most significant, affecting potential socioeconomic health differences.

Originality and significance: if not novel, please include reference

The study's results are crucial given all the (European) developments in lung cancer screening and the launch of pilots and

national programs in many countries throughout the world. Despite ample scientific evidence for the effectiveness of lung cancer screening, high-quality implementation requires significant efforts and faces several challenges. With these data, other countries can benefit of all experiences of this UK programme.

With a detection rate of 200 lung cancers per month where about 20-24% of the lung cancer deaths can be possibly prevented indicates the enormous potential impact of this programme on public health, potentially reducing socio-economic health differences (as supported by figure 5).

Data & methodology: validity of approach, quality of data, quality of presentation

With the “real world environment” data, the authors used the best data as is available in such a national screening programme, recognizing also the potential limitations of the data sources used. A longer follow-up will allow the authors to further evaluate the programme. Nevertheless, the current data are an important contribution to current evidence in lung cancer screening.

Appropriate use of statistics and treatment of uncertainties

The statistics as described are appropriate.

Conclusions: robustness, validity, reliability

The manuscript conclusions are robust, valid and reliable. However, more details are needed for a more detailed/precise (cost-)effectiveness analyses.

Suggested improvements: experiments, data for possible revision

Although the authors described this programme detailed, some additions might further improve the manuscript. For example, the stages are presented, which showed a clear stage shift that would be a requirement for further impact on lung cancer mortality and all-cause mortality. More details about histology and sex differences can provide further evidence about what lung cancers are detected most and potential sex-differences in lung cancer screening.

Aspects as recruitment, screening numbers and lung cancer detection have been presented. And although these details are already very informative and interesting, more details about the screening test results, test positives, false positives e.g. are missing. These data are important key performance indicators of a programme.

Furthermore, many people had at least one IF reported. Important for a programmas' (cost)effectiveness is which IFs are clinical relevant and requires further diagnostic work-up and treatment. Some IFs might also already known/under treatment, while others are newly detected in (a)symptomatic individuals? Adding some additional details about the management of IFs among the participants will support the readers understanding of the impact of IFs in a lung cancer screening programme. The impact can be both negatively as well as positively, as discussed in the discussion section.

The flowchart as presented in figure 1 can be improved by providing more details and by indicating the numbers more precise. For example, 250,329 people underwent follow-up screening, followed by a detection rate of 1.4%. The box related to a 12 months follow-up is linked to the box about the detection rate. But what about the 3-months and 12-months follow-up? and 24/48/72 months follow-up? How many people were referred because of a positive screening test result? More than 30% received a 3 months of for surveillance LDCT scans. Then, another 14% received a 12-months recall for such a scan. This seems to be higher than expected from other trials that using volumetric CT screening with three possible outcomes? Furthermore, in the figure, it is also unknown how many people were selected for the LHC by what recruitment method.

References: appropriate credit to previous work?

Sufficient use of references.

Clarity and context: lucidity of abstract/summary, appropriateness of abstract, introduction and conclusions

The abstract can be improved by providing some more details in the methods and results section. Otherwise, it is a missed opportunity not to elaborate more explicitly on such important results:

The part “The programme adheres to a unified protocol, with rigorous quality assurance, delivered through regionally federated clinical infrastructure and leadership, embedded within national strategic, clinical, and economic frameworks...” are more general statements, although some more details about the programme can improve the function of the abstract. Some reflection on the aim to present the progress and outcomes of the NHSE LCS programme (as stated as aim) might be helpful.

The part “...Numerous countries are actively exploring the implementation of lung cancer screening in response to the accumulated evidence for reduction in mortality. This manuscript offers a rich source of information on a national programme that follows a single protocol, paving the way for many other proposed programmes internationally. The programme has demonstrated feasibility and scalability in reaching high-risk and underserved populations. ...” is more related to a conclusion.

In the introduction, lines 74-85 have some overlap that can be removed to improve the readiness of this part.

There seems to be a typo at line 312: “...August 202, cancer ...” 202 should be 2023 I assume? (as also stated in line 104 of the Supplementary Methods)

(Remarks on code availability)

Reviewer #3

(Remarks to the Author)

Implementation of a National Lung Cancer Screening Programme: The UK National data at 5 years

Abstract

"The data and methodology have informed the UK National Screening Committee's recommendations, are referenced in Lord Darzi's Report"

Most readers outside the UK will have no knowledge of Lord Darzi's Report.

It would have been helpful to have had pages numbered.

"Quality Assurance standards prepared for the lung cancer screening programme" were revised in 2025, but without substantial change to the programme.⁹

Ref 9. NHS England. Standard protocol and quality assurance standards for the Lung Cancer Screening Programme. 2025. It would be nice to provide a link to the standards.

"These potential participants are contacted by letter, SMS, and/or telephone and invited to a lung health check (LHC) to assess their risk of developing lung cancer."

I was under the impression that the initial Pilot name "Targeted Lung Health Check" omitted the word "cancer" to avoid negative connotations. But now the national Lung Cancer Screening Programme uses the word "cancer". Other jurisdictions, such as Ontario, use "Ontario Lung Screening Program". Why the change of thinking?

"All screening LDCT scans are reported by thoracic radiologists with lung cancer expertise who regularly participate in lung cancer MDTs,"

MDT is undefined in manuscript.

In the Methods it was not explicitly stated whether nodule risk was determined using volumetric approaches or not. What defines an indeterminate result?

"Cancer detection rates are per participant"

This seems an odd denominator, as cancer detection rates are based on samples of well more than one. And for any given person they either get cancer coded 1 or do not get cancer coded 0. And what is really presented for cancer detection rate is a proportion estimated on a sample, not a single individual.

"Rate" in epidemiology usually means that a period of time is specified, but this is sometimes loosely used.

"Diagnoses considered LCS attributable were those that followed a CT scan or an LHC resulting in a high-risk assessment and were diagnosed within 185 days of the above event (see supplementary methods)."

This seems like a short window for defining a screen-detected lung cancer. I include a figure of time to diagnosis for screening LungRADS 4's at the Lahey Hospital System. Overall, about 20% of diagnoses were after 6 months. Where did the 185 days come from? It might also be helpful to explain that the 185 days would exclude indeterminate or equivalently LungRADS 3 scans.

I don't think that IMD has been defined before use.

"All demographic variables were included in each regression model and due to the high number of missing demographic data recorded in the pilot sample, separate categories were retained for age, ethnicity, or deprivation not known.

Why do you have missing age? And are you analyzing age as categories not continuous. This leads to a loss of information. Treating missing data as an extra category or the "missing indicator method" has been criticized for many reasons. It is conceptually wrong and can bias estimates. Imputation is preferred.

What were the numbers and proportions missing?

"2.3% (13,231 of 528,686) met the risk threshold but were ineligible for LDCT on the basis of exclusion criteria."

I do not recall seeing the exclusion criteria described?

"Figures S1 and S2 summarise programme geography and proportional national roll-out as a marker of coverage, by cancer alliance."

I forgot or was unclear as to what "cancer alliance" is and does?

"but women were less likely to undergo a LDCT scan as a proportion of those attending a LHC (48.4% vs. 56.7%; OR 0.72 CI 0.71-0.73, p<0.001)."

Why the lower scan rate in women? This may be important, especially given their higher CDR.

"A smaller proportion of participants eligible and invited for a LHC in the 'other' ethnic group attended than those of 'white' ethnic group (18,295/97,265, 18.8% vs 84,765/295,450, 62.5%; OR 0.09 (CI 0.09- 0.10 p<0.001))."

This is a huge difference.

“Ethnicity data was not known for 32.6% of the 582,700 individuals eligible for a LHC.”
Data “were”, here and elsewhere.

“Ethnicity data was not known for 32.6% of the 582,700 individuals eligible for a LHC. Participants attending a LHC who were of other ethnicities, were less likely to attend a LDCT than LHC attendees of white ethnicity (9,320/18,295, 51.0%, vs.95,060/184,765, 51.5%; OR 0.75 (CI 0.73-0.78, P<0.001)).”

Note that there is a rounding error in the 51.5%.
 $95060 / 184765 = .51449138 = 51.4\%$, not 51.5%

Also, the OR, Cis and P-value appear wrong. See STATA analysis printout of the numbers provided below:

```
. cci 9320 8975 95060 89705
Proportion
| Exposed Unexposed | Total exposed
-----+-----+-----
Cases | 9320 8975 | 18295 0.5094
Controls | 95060 89705 | 184765 0.5145
-----+-----+-----
Total | 104380 98680 | 203060 0.5140
| |
| Point estimate | [95% conf. interval]
|-----+-----|
Odds ratio | .9799418 | .950498 1.010298 (exact)
Prev. frac. ex. | .0200582 | -.0102978 .049502 (exact)
Prev. frac. pop | .0103198 |
+-----+-----+
chi2(1) = 1.71 Pr>chi2 = 0.1913
```

“However, 37.7.0% (5,665/15,050) of LHC participants from the least deprived areas underwent LDCT scanning vs 57.2% (48,7600/85,285) people from quintile 1 (OR 0.39 CI 0.37-0.40 p<0.001).”

Some readers may jump to the conclusion that the difference in scans may be due to personal refusal to accept the offer of scan. But is it not possible or likely that risk was higher in the more deprived peoples? Can the readers’ interpretation be guided here?

“Cancer outcomes were censored in August 202,cancer rates being increased by the LDCT censor date (see supplementary methods).”

The idea in the last part of this sentence is unclear.
Note the year “202” and missing space are the authors.

“The programme is impacting national statistics for lung cancer where early stage detection rates are now well above pre-pandemic levels.”

Detection rates of early-stage lung cancer can be confounded by over-diagnosis, lead-time bias and length time bias, so are not good metrics to use. A drop in the incidence rate of advanced cancers is a better and more convincing metric to present.

“Participation rates, which compare favorably to other international experience, are evidence of strong public engagement, ...”

Can the “participation rates” be neatly summarized in one or a few statistics? I found I had to go searching and digging through tables to get a sense of it.

What about program sensitivity, specificity, false positive rates, interval cancer rates, harms done, and ... ?

There seem to be a lot of LCS program quality indicators that are not presented. They could be put into tables in supplement if space requires.

“Future longitudinal comparisons of socioeconomic group-stratified stage distributions between lung cancers diagnosed through LCS and those presenting clinically should help clarify whether stage shift is being achieved in all groups or a specific socioeconomic group.”

The discussion in the manuscript leaves the impression that determining successful screening will be done through evaluation of stage shift to early disease. As mentioned before this is a poor metric. See Feng X, Zahed H, Onwuka J, Callister MEJ, Johansson M, Etzioni R, et al. Cancer Stage Compared With Mortality as End Points in Randomized Clinical Trials of Cancer Screening: A Systematic Review and Meta-Analysis. JAMA. 2024;331(22):1910–7.

“Lung screening must meet the standards set for screening in the NHS, with focus on; quality assurance and performance; better data systems bespoke to the programme; and a more engaged research community to the develop the future programme improvements.”
Awkward language/grammar.

“A further limitation is that more detailed participant record-level data with cancer diagnoses linked from national registration data are only available for the initial phases of the programme.”

This seems like an important limitation and is hard to understand. Why cannot cancers in the Registry not consistently be linked to screened participants over time?

Figure 1

From text

“Targeted Lung Health Check (TLHC) Programme”

“invited to a lung health check (LHC)”

Use of TLHC versus LHC suggests that they mean different things and the former is for screening? And the later is for health and risk assessment to see if screening should be offered.

If his interpretation is correct, then in Figure 1, the first box to branch off to the right says “Did not attend TLHC”. This branching off appears to come to early because the individuals have not yet had a LHC and an offer to attend TLHC (screening).

There is one arrow leading to “Lung cancer diagnosed” coming from “At 12 months”. Arrows should be coming from all three boxes in that row.

Figure 2. Some of the fonts are too small to be easily read. The data points on the figure lines can be enlarged, and number of decimals can be reduced.

Table 1. Acronyms are not explained.

Figure 4. “(2) LDCT attendance of those who attended a LHC”

Should it not be “LDCT attendance of those who attended an LHC and were found to be at high risk and were offered lung screening”. If you are just looking at screening attendance in all who attended LHC, you are conflating eligibility and attendance.

Figure 5. Not only is it questionable practice to look at early-stage cancer as a metric because of potential biases, such as resulting from over-diagnoses, using proportion is additionally poor practice, because a change in any one level of stage will lead to change in proportions of the other levels, even when the incidence rates remain unchanged. Ideally, monitoring changes in incidence rates of stage IV cancer is preferred.

Figure 6 is a Table.

The Supplement needs a Table of Content and Headings.

(Remarks on code availability)

No statement about code was made or code presented. But the coding involved was only for simple descriptive statistics and logistic regression models using standard R packages. Not much to worry about here.

No statement was made about data availability.

Version 1:

Reviewer comments:

Reviewer #1

(Remarks to the Author)

This Reviewer congratulates the authors of the manuscript in their efforts to respond to the recommendations which I have made and also those from the other Reviewers. Practically all have been taken on board and incorporated into the text.

This manuscript now not only provides a comprehensive and detailed account of the UK’s Implementation of a National Lung Cancer Screening Programme (NLCSP), over the last five years, but will also be a very useful document for international groups starting or implementing LCS programmes.

The Standard Protocol and the QAS addition to the supplementary section is also a bonus for the external readers.

One specific comment on re-reading the manuscript, I question the inclusion of Table 2, which provides information from the initial phase on Incidental Findings (IF). This table provides data on the 77,182 IF from 114,430 individuals who had a LDCT in the initial phase. However, this 114,430 only represents 14.6% of the total number of individuals (528,684) who underwent an initial LDCT in the reported programme. As this is such a small proportion of the total number of individuals recruited into the current programme, is it a true reflection of what is happening? Would it not be wise to hold back on this IF data for another publication when the data was much more mature ?

Signed

Confidential Information Redacted

(Remarks on code availability)

Reviewer #2

(Remarks to the Author)

The National Lung Cancer Screening Programme presents significant and valuable data regarding the implementation of lung cancer screening. This practice is gaining traction globally, particularly in European countries. The manuscript makes a noteworthy contribution to the body of evidence surrounding the implementation of lung cancer screening, the impact, but also the challenges in implementation.

The authors have made substantial revisions to the manuscript, providing the reader with a more detailed understanding based on the available data. As future data becomes accessible, we can anticipate additional evidence on relevant key performance indicators, although it is too early to draw conclusions at this stage.

(Remarks on code availability)

Reviewer #3

(Remarks to the Author)

I have read the authors' responses to all reviewers' comments and the revised manuscript with changes made. The authors for the most part have responded adequately to the reviewers' comments. In several places where presentation had shortcomings, it was due to lack of required data at this time, which to an extent will be overcome in the future.

AUTHORS' RESPONSE to REVIEWER 2's COMMENT: "Cancer diagnoses were derived from the cancer registration dataset rather than an individual patient record."

If this is the case, then how were interval cancers identified?

I see from reading later on that data to calculate interval cancers accurately were not available.

"An independent review of the NHS in England in 2024, highlighted the improvement in early stage cancer detection "... likely to be in significant measure due to the Targeted Lung Health Check programme...".¹²"

I'm glad there is a space between "in" and "significant".

"Cancer detection is expressed as a proportion of the prevalence round LDCTs (equivalent to 'per participant')."

I am fine with this presentation if the authors prefer it. However, there is no reason why not to present it as per cent, that is per 100 individuals. Generally, people have a clear interpretation of percent, but possibly less so of a straight probability.

Small thing: There are many spacing errors where the authors have two and even three spaces between sentences. I presume that the journal will clean up the presentation.

Confidential Information Redacted

(Remarks on code availability)

Code was not presented. Although the datasets for England are understandable complex, the code to analyze the simple descriptive epidemiological statistics should be simple and easily figured out.

LCS Nature Medicine Paper Peer review Responses

Reviewer #1 (Remarks to the Author):

The authors provided a comprehensive and detailed account of the UK's Implementation of a National Lung Cancer Screening Programme (NLCSP) over the last five years. The success is seen in the actual numbers; over 2.5M individuals invited, 49% of whom attended an appointment (face to face or by telephone); over 500K have had their first LDCT scan and 7,193 lung cancers (1.4% cancer detection rate) have been identified by March 2025.

Thus, the NLCSP is an excellent example of successful implementation of lung cancer LDCT screening in the real world and is a credit to all the individuals involved, but particularly the individuals at the coal-face who have delivered the service on a daily basis, listed under the 'Lung Cancer Screening Programme Research Consortium'.

Response: Thank you for acknowledging the hard work of the team and the rigour of the work and data presented by a wide, equitable team.

This article should not, however, solely be a report of a success story, as many of the international readers of this article will be setting up implementation programmes in their own countries or in the midst of feasibility studies.

Disappointingly, the authors have missed an opportunity to provide this guidance, which would significantly increase the value of this manuscript and make it more informative for an international readership.

Response: Thank you for this insightful feedback and specific recommendations provided below. We have addressed these as well as expanding other areas of the manuscript with methodology for shared learning

Furthermore, given a number of questions that are already covered by the Standard Protocol and The Quality Assurance Standard, we have now included these in the supplementary appendix.

The following points need further elaboration within the manuscript.

- Starting at the beginning – what evidence was utilised to even start the TLHC? The UKLS trial was the first RCT to utilise a lung cancer risk model to select high risk participants, this was followed by five local implementation projects (referenced in the manuscript); all of these implementation projects used either one or both of the two validated risk models (LLPv2 and or PLCO m2012). The UK National Screening committee (UKNSC) commissioned the cost effectiveness analysis of lung cancer

screening, and it was only because of the use of risk models to recruit high risk individuals, that lung cancer screening was shown to be cost effective, <https://view-health-screening-recommendations.service.gov.uk/lung-cancer/> This publication provides details of the analysis of the UK cost-effectiveness modelling, which resulted in the UKNSC recommending LDCT screening in targeted high-risk populations in 2022 and the national lung cancer screening was supported by the UK Prime Minister in 2023.

Unless this pivotal initial step is addressed as outlined above, those involved in the development of screening programmes will not have this crucial data as a basis for debating with their Ministry of Health, the necessary focus on targeted risk groups which are cost effective.

Response: Thank you for highlighting the value of describing the evolution of the program. This has been addressed in the introduction, including a reference to the health economics evaluation commissioned by the UKNSC.

- The Stage 1 & II data provided in this manuscript is in the same ballpark as the UKLS trial and the five UK implementation projects, which is the best parameter of success to date (Fig 5), until the NHS LCS mortality data is available.

Response: Thank you for noting this metric of success – the reference to UK pilots added to existing trial data references.

- There needs to be an explanation as to why the NHS LCS programme is run on a 24 month repeat basis, instead of 12 months, as the majority of the international RCTs were undertaken on a 12 month repeat basis. Only the NELSON trial included a two -year interval in their trial.

Response: Thank you, this has now been explained in some detail in the methods section and references added.

- The authors mention the issues around Incidental Findings (IF) and the impact on Primary care. The potential of utilising coronary artery calcification (CAC) as well as Emphysema / COPD data, is mentioned but needs much greater clarity in respect of what is proposed in the future (noted current status in Annex 2 in Ref 9). Internationally, IF is considered a major issue particularly in respect of how it will impact on their primary care providers. How will this be funded in the UK national screening programme and is it cost effective?

Response: As you rightly highlight, the effective management of incidental findings is critical for success of any screening programme. The program is supported by an incidental findings protocol that includes recommended management, legal

considerations informed by two separate legal teams and a national management pathway to ensure that required actions for IFs are completed. The QA Standard, where the IF protocol appears as an appendix, is now available as online supplementary material. Whilst it is not possible to capture the content of the IF protocol, we have added more detail and signposted to the online material.

We anticipate in future that data on IF's may include patient level prescription details and cardiac or respiratory outcomes from primary care records, and hospital visits for cardiorespiratory and other specialist services. Additionally, tracking the reporting of such conditions nationally and national statistics such as HES, will help to ensure consistent quality of reporting and referrals to inform where deeper dive analyses should occur. We have elaborated upon this in the manuscript. Further examination of the data is out of scope for this data unfortunately, and awaiting its preparation was not a justifiable reason to delay release of this main programme data. We intend to bring the above data back in a future manuscript.

The manuscript is also updated to reference Annex 3 of Ref 9 which gives recommended management pathways for incidental findings. Both the Standard Protocol and the QA Standard are now included as supplementary online material for convenience.

- Lung cancer screening at a national level, impacts on all of the workforce in the Lung Cancer screening pathway, however, the major impact is on the Radiology workforce. Are the current arrangements sustainable in the UK (some description should be included on the NHS/ private provision of radiology reporting), as well as the current use and future potential of incorporating artificial intelligence (AI) platforms in the reporting pathway. The NHS LSC programme is dependent on using mobile CT units, is this sustainable moving into a long-term national programme ?

Response: Thank you, the manuscript describes outsourced capacity, for lung health check call provision, and LDCT scanning and reporting. We have incorporated the recommended points above into the manuscript discussion.

- The discussion point: ' better data systems bespoke to the programme' .

The UK Biobank has provided international leadership in recruitment, collection of questionnaire data, clinical findings, together with data on a wide range biomarkers; providing an invaluable resource for future clinical research. On reflection, should the TLHC/ NHS LCS programme not have considered utilising an integrated database with a central hub for the CT images, instead of the disparate system throughout the country? Can this situation be retrieved to improve access to future data and the CT images by the research community?

Response: Thank you for requesting this. This has been a priority in the planning process and was specifically defined in the Protocol and QA standard. However, the track record of NHS IT systems, and the speed with which the evaluation was required meant that a national system was not prioritised. Presently focus has been on providing the best local data provisioning possible, supported by national data teams, with active development of a nation system in progress.

- Finally, the NHS LCS programme has succeeded in recruiting ~ 50% of the invited population, which is lower than other cancer screening programmes in the UK (acknowledge it's just starting) . However, the 50% which do not participate are mainly within the lower socio-economic deprived communities, who have a higher incidence of lung cancer, potentially as high as 2.5%. Surely this has to be considered and discussed in this manuscript?

Response: Thank you for highlighting this important point which is held in high regard by the authors and has been the focus on innovation and research in the programme. The data available that informs this inequity is described in the discussion. In view of this not being clear, an additional comment has been made on the point explicitly made above related to socio-economic deprivation and benefit from the programme.

Overall, this manuscript heralds the success story of Lung Cancer Screening in the UK but fails to provide a blueprint for screening in other countries and fails to address future challenges.

Response: Thank you for this comment. It is well taken, and we have addressed this by attending to the reviewers comments and now including the Protocol and QA Standard as online supplements so that the reader may go direct to the blueprint. This is freely available and adaptable according to local considerations – a point that we have made at the end of the discussion.

Other comments

References incomplete:

9. England. Standard protocol and quality assurance standards for the Lung Cancer Screening Programme. 2025.

Response: Updated.....

15. NHS England. Implementing a timed lung cancer diagnostic pathway. 2024.

Response: Updated.....

Does Figure S3 actually add anything to the manuscript?

Response: This was debated at time of manuscript writing. In view of the feedback, we have removed this.

Reviewer #1 (Remarks on code availability):

Reviewed the information on https://github.com/craig-parylo/610_tlhc

The information provided would require an individual with the necessary computer /IT experience in order to answer the question - are the results reproducible.
Recommend that you have a Reviewer with these IT skills to answer this question.

Reviewer #2 (Remarks to the Author):

Summary of the key results

The authors make an exceptional contribution to scientific developments in the field of (the implementation of) lung cancer screening by describing the first results of the national lung cancer screening programme as implemented in the UK. Although lung cancer screening is proven effective in reducing lung cancer/all-cause mortality, many challenges remain in the implementation of nationwide programmes.

For this programme, more than 2 million individuals were invited for a lung health check, which is according to the authors almost one third of the estimated eligible population for lung cancer screening in the UK. Over 7,000 lung cancers have been diagnosed, with a cancer detection rate of 1.4% of participants undergoing baseline scans. A favorable stage distribution towards early stages could be confirmed and are comparable with results of other lung cancer screening trials.

This paper also highlights other important aspects of lung cancer screening programme, such as the risk-based recruitment and screening uptake, with special attention to people living in the more deprived areas, where lung cancer incidence is relatively higher and where the impact of lung cancer screening might be most significant, affecting potential socioeconomic health differences.

Response: Thank you. This was achieved through careful planning and the hard work of teams at national and local level.

Originality and significance: if not novel, please include reference

The study's results are crucial given all the (European) developments in lung cancer screening and the launch of pilots and national programs in many countries throughout

the world. Despite ample scientific evidence for the effectiveness of lung cancer screening, high-quality implementation requires significant efforts and faces several challenges. With these data, other countries can benefit of all experiences of this UK programme.

With a detection rate of 200 lung cancers per month where about 20-24% of the lung cancer deaths can be possibly prevented indicates the enormous potential impact of this programme on public health, potentially reducing socio-economic health differences (as supported by figure 5).

Response: Thank you. We have tracked these metrics carefully to ensure rapid and effective roll-out of the programme that we hope to scale up rapidly.

Data & methodology: validity of approach, quality of data, quality of presentation
With the "real world environment" data, the authors used the best data as is available in such a national screening programme, recognizing also the potential limitations of the data sources used. A longer follow-up will allow the authors to further evaluate the programme. Nevertheless, the current data are an important contribution to current evidence in lung cancer screening.

Response: Thank you – we viewed it as important to share the data we have access to and that has been requested widely from the international lung cancer and screening communities. Publishing these data would reinforce the opportunity described and provide visibility with which to advocate for further implementation and research in this area.

Appropriate use of statistics and treatment of uncertainties
The statistics as described are appropriate.

Conclusions: robustness, validity, reliability
The manuscript conclusions are robust, valid and reliable. However, more details are needed for a more detailed/precise (cost-)effectiveness analyses.

Response: Thank you, we appreciate the importance of this comment. We reference the detailed health economic evaluation commissioned by the UK NSC, where screening program data were included in some of the model inputs. The HE analysis was completed in November 2022 when the pre-audit final results were made available and the link is on the UKNSC website immediately beneath the recommendation. There is a final publication following external audit that is currently in press. This is all addressed in the introduction.

Suggested improvements: experiments, data for possible revision
Although the authors described this programme detailed, some additions might further

improve the manuscript. For example, the stages are presented, which showed a clear stage shift that would be a requirement for further impact on lung cancer mortality and all-cause mortality. More details about histology and sex differences can provide further evidence about what lung cancers are detected most and potential sex-differences in lung cancer screening.

Response: Thank you for these comments, which we recognise as key indicators. Literature review of key existing studies in this area suggest a longer follow up time will be needed to confidently assess impact on mortality – we have annotated the discussion to note this (e.g. Grannis et al). This will be an important focus of future work. We have presented sex and ethnicity data in lung cancers as currently available. We await further patient-level data linked to outcomes to explore this further. It is anticipated that resources such as those generated by the DART study (Data Using ARtificial InTelligence to Improve Patient Outcomes with Thoracic Diseases, REC ref: 21/WM/0278) will enable this.

Aspects as recruitment, screening numbers and lung cancer detection have been presented. And although these details are already very informative and interesting, more details about the screening test results, test positives, false positives e.g. are missing. These data are important key performance indicators of a programme.

Response: Early data on CT scan level outcomes are presented which gives rates of cancer detection by baseline, and 3 vs 12 month surveillance CT scans for indeterminate screening outcomes.

We agree that the additional data the reviewer highlights are important to monitor the program. We have added additional data where available and commented about the importance of this in the discussion, saying that data such as recall for surveillance on IFs are monitored by some consortia radiology providers but the key indicators, as defined by our draft effectiveness standards are not yet reliably collected.

Furthermore, many people had at least one IF reported. Important for a programmes' (cost)effectiveness is which IFs are clinical relevant and requires further diagnostic work-up and treatment. Some IFs might also already known/under treatment, while others are newly detected in (a)symptomatic individuals? Adding some additional details about the management of IFs among the participants will support the readers understanding of the impact of IFs in a lung cancer screening programme. The impact can be both negatively as well as positively, as discussed in the discussion section.

Response: Thank you – we have elaborated further on this in the discussion and reference the annex 2 and 3 that give detailed instructions as to the management of IFs – downstream management is previously described in the literature and will likely be available from future regional data. Both the Standard Protocol and the QA Standard are now included as supplementary online material for convenience. The latter contains an

appendix that details the management of IFs in the program including nationally agreed pathways of care.

The flowchart as presented in figure 1 can be improved by providing more details and by indicating the numbers more precise. For example, 250,329 people underwent follow-up screening, followed by a detection rate of 1.4%. The box related to a 12 months follow-up is linked to the box about the detection rate. But what about the 3-months and 12-months follow-up? and 24/48/72 months follow-up? How many people were referred because of a positive screening test result? More than 30% received a 3 months of for surveillance LDCT scans. Then, another 14% received a 12-months recall for such a scan. This seems to be higher than expected from other trials that using volumetric CT screening with three possible outcomes?

Furthermore, in the figure, it is also unknown how many people were selected for the LHC by what recruitment method.

Thank you for raising that the flow chart is not clear. We have updated the chart to make it clear that the cancer detection quoted is the total from any screening CT expressed as a proportion of the participants. Cancer diagnoses were derived from the cancer registration dataset rather than an individual patient record. The diagram has been amended to avoid confusion about surveillance CTs and incidence rounds, showing the surveillance CT proportion is in keeping with the literature. We did not collect the referral rates as part of this study, but these data are to be collected and used as part of the effectiveness standards that are currently developed and awaiting stakeholder approval. We have included an explanation of this in the discussion.

References: appropriate credit to previous work?

Sufficient use of references.

Clarity and context: lucidity of abstract/summary, appropriateness of abstract, introduction and conclusions

The abstract can be improved by providing some more details in the methods and results section. Otherwise, it is a missed opportunity not to elaborate more explicitly on such important results:

The part "The programme adheres to a unified protocol, with rigorous quality assurance, delivered through regionally federated clinical infrastructure and leadership, embedded within national strategic, clinical, and economic frameworks..." are more general statements, although some more details about the programme can improve the function of the abstract. Some reflection on the aim to present the progress and outcomes of the NHSE LCS programme (as stated as aim) might be helpful.

The part "...Numerous countries are actively exploring the implementation of lung cancer screening in response to the accumulated evidence for reduction in mortality. This manuscript offers a rich source of information on a national programme that follows a single protocol, paving the way for many other proposed programmes internationally. The programme has demonstrated feasibility and scalability in reaching high-risk and underserved populations. ..." is more related to a conclusion. In the introduction, lines 74-85 have some overlap that can be removed to improve the readiness of this part.

Response: Thank you for these suggestions – we have elaborated on the methods and incorporated these comments and edited the abstract as suggested. Editorial advice previously has been to whet the reader's appetite with a signal to the headline results without presenting all of the major data. In view of the limited word count here, we have kept it very focused. We would be happy to review this if wished by the editors, and peer reviewers. We have removed some duplication regarding NHSE and the programme name.

There seems to be a typo at line 312: "...August 202, cancer ..." 202 should be 2023 I assume? (as also stated in line 104 of the Supplementary Methods)

Response: Thank you for pointing out this typo error

Reviewer #3 (Remarks to the Author):

Implementation of a National Lung Cancer Screening Programme: The UK National data at 5 years

Abstract

"The data and methodology have informed the UK National Screening Committee's recommendations, are referenced in Lord Darzi's Report"

Most readers outside the UK will have no knowledge of Lord Darzi's Report.

Response: Thank you for pointing this out – we have edited this.

It would have been helpful to have had pages numbered.

Response: Thank you for pointing out this omission

"Quality Assurance standards prepared for the lung cancer screening programme" were revised in 2025, but without substantial change to the programme.⁹

Ref 9. NHS England. Standard protocol and quality assurance standards for the Lung Cancer Screening Programme. 2025.

It would be nice to provide a link to the standards.

Response: Thank you for pointing out this omission which we have added, and also provided the document as supplementary material.

"These potential participants are contacted by letter, SMS, and/or telephone and invited to a lung health check (LHC) to assess their risk of developing lung cancer."

I was under the impression that the initial Pilot name "Targeted Lung Health Check" omitted the word "cancer" to avoid negative connotations. But now the national Lung Cancer Screening Programme uses the word "cancer". Other jurisdictions, such as Ontario, use "Ontario Lung Screening Program". Why the change of thinking?

Response: Thank you. The programme name has been a point of careful focus throughout the evolving programme. NHSE went through a consultation process to understand participant perception of the programme name. This included undertaking some behavioural science led assessment of whether including the word 'cancer' was preferred. It was found that NOT including the word cancer led to people being unclear what the focus of the programme was and made people less willing to attend. We have added this detail to the discussion as personal correspondence.

"All screening LDCT scans are reported by thoracic radiologists with lung cancer expertise who regularly participate in lung cancer MDTs,"
MDT is undefined in manuscript.

Response – Thank you – this has been expanded

In the Methods it was not explicitly stated whether nodule risk was determined using volumetric approaches or not. What defines an indeterminate result?

Response – Thank you. Volumetry and computer aided detection is mandated – we have added this. British Thoracic Society guidelines are followed for lung nodule management. We have added some information, but further details are in the Standard Protocol, now included in supplementary material.

"Cancer detection rates are per participant"

This seems an odd denominator, as cancer detection rates are based on samples of well more than one. And for any given person they either get cancer coded 1 or do not get cancer coded 0. And what is really presented for cancer detection rate is a proportion estimated on a sample, not a single individual.

"Rate" in epidemiology usually means that a period of time is specified, but this is sometimes loosely used.

Response: Thank you for this observation. We have clarified the text. The number of cancers detected is expressed with the denominator as the number of baseline scans, equivalent to the participant. We were not able to pull out which cancers were detected

on incidence round CTs, although the vast majority at this time point were from the baseline or following surveillance CTs. Figure 1 shows that 25% of participants had had an incidence round CT. It is appreciated that this may increase the proportion of cancers detected and this has been added in the discussion.

“Diagnoses considered LCS attributable were those that followed a CT scan or an LHC resulting in a high-risk assessment and were diagnosed within 185 days of the above event (see supplementary methods).”

This seems like a short window for defining a screen-detected lung cancer. I include a figure of time to diagnosis for screening LungRADS 4's at the Lahey Hospital System. Overall, about 20% of diagnoses were after 6 months. Where did the 185 days come from? It might also be helpful to explain that the 185 days would exclude indeterminate or equivalently LungRADS 3 scans.

Response: Thank you for highlighting this. This is defined in the supplementary methods. Where a nodule surveillance scan is undertaken, this would be the scan from which the 185 day period starts, so any nodule in which surveillance was recommended before biopsy would not be miscounted as outside the 185 day limit. The decision to use the 185-day limit as a classification method was based on it representing the maximum acceptable time between a person's lung cancer screening (LCS) scan and subsequent diagnosis to consider the two events as directly linked. This means we can confidently attribute a lung cancer diagnosis to the screening process if it occurred within 185 days of the scan. Our method for establishing this limit involved survival analysis to assess the likelihood of diagnosis following a scan and time-to-event segmentation. This process was quality assured by a member of the National Cancer Registration and Analysis System team. See below, this is included in supplementary material.

Redacted

I don't think that IMD has been defined before use.

Response: Thank you. Corrected

"All demographic variables were included in each regression model and due to the high number of missing demographic data recorded in the pilot sample, separate categories were retained for age, ethnicity, or deprivation not known.

Why do you have missing age? And are you analyzing age as categories not continuous. This leads to a loss of information.

Treating missing data as an extra category or the "missing indicator method" has been

criticized for many reasons. It is conceptually wrong and can bias estimates. Imputation is preferred.

What were the numbers and proportions missing?

Response: Thank you for this comment. Data were obtained directly from primary care electronic records where data are incomplete, including age and sex. Table 1 shows that the missing proportion ranged from 1 to 33%. For high levels of missingness we followed the view that imputation could be misleading and instead elected to present this real-world data as it was found¹. We agree that age categorisation will lose some data, but the purpose was to show what can be expected within age categories commonly quoted in analyses of real-world data, reflecting age of employment and retirement.

"2.3% (13,231 of 528,686) met the risk threshold but were ineligible for LDCT on the basis of exclusion criteria."

I do not recall seeing the exclusion criteria described?

Response: For brevity, exclusion criteria are described in the Standard Protocol (now included in the supplementary material).

"Figures S1 and S2 summarise programme geography and proportional national roll-out as a marker of coverage, by cancer alliance."

I forgot or was unclear as to what "cancer alliance" is and does?

Response: Thank you – we have added a definition.

"but women were less likely to undergo a LDCT scan as a proportion of those attending a LHC (48.4% vs. 56.7%; OR 0.72 CI 0.71-0.73, $p < 0.001$)."

Why the lower scan rate in women? This may be important, especially given their higher CDR.

Response: This is an important observation and one that has been observed in other studies including UKLS. We have added a comment in the discussion

"A smaller proportion of participants eligible and invited for a LHC in the 'other' ethnic group attended than those of 'white' ethnic group (18,295/97,265, 18.8% vs 84,765/295,450, 62.5%; OR 0.09 (CI 0.09- 0.10 $p < 0.001$))."

This is a huge difference.

Response: We agree and this needs to be addressed in the ongoing program. At least we have quantified the problem. We have added further text in the discussion.

"Ethnicity data was not known for 32.6% of the 582,700 individuals eligible for a LHC." Data "were", here and elsewhere.

Response: Thank you – corrected here, and elsewhere.

"Ethnicity data was not known for 32.6% of the 582,700 individuals eligible for a LHC. Participants attending a LHC who were of other ethnicities, were less likely to attend a LDCT than LHC attendees of white ethnicity (9,320/18,295, 51.0%, vs.95,060/184,765, 51.5%; OR 0.75 (CI 0.73-0.78, P<0.001))."

Note that there is a rounding error in the 51.5%.

$95060 / 184765 = .51449138 = 51.4\%$, not 51.5%

Also, the OR, Cis and P-value appear wrong. See STATA analysis printout of the numbers provided below:

Response: Thank you, this has been corrected. The ORs are adjusted for all of the other demographics. Please also note that some of the values in table 1 do not correspond exactly to the figures. The samples the regressions were run on were slightly different samples, due to missing data in people who attended both a LHC and scan.

```
cci 9320 8975 95060 89705
Proportion
| Exposed Unexposed | Total exposed
-----+-----+-----
Cases | 9320 8975 | 18295 0.5094
Controls | 95060 89705 | 184765 0.5145
-----+-----+-----
Total | 104380 98680 | 203060 0.5140
||
| Point estimate | [95% conf. interval]
|-----+-----
Odds ratio | .9799418 | .950498 1.010298 (exact)
Prev. frac. ex. | .0200582 | -.0102978 .049502 (exact)
Prev. frac. pop | .0103198 |
+-----+-----
chi2(1) = 1.71 Pr>chi2 = 0.1913
```

"However, 37.7.0% (5,665/15,050) of LHC participants from the least deprived areas underwent LDCT scanning vs 57.2% (48,7600/85,285) people from quintile 1 (OR 0.39 CI 0.37-0.40 p<0.001)."

Some readers may jump to the conclusion that the difference in scans may be due to personal refusal to accept the offer of scan. But is it not possible or likely that risk was higher in the more deprived peoples? Can the readers' interpretation be guided here?

"Cancer outcomes were censored in August 2023, cancer rates being increased by the LDCT censor date (see supplementary methods)."

The idea in the last part of this sentence is unclear.

Note the year "202" and missing space are the authors.

Response: We agree that the explanation you offer is the likely reason for the difference observed and we make this point in the discussion (page 14).

"The programme is impacting national statistics for lung cancer where early stage detection rates are now well above pre-pandemic levels."

Detection rates of early-stage lung cancer can be confounded by over-diagnosis, lead-time bias and length time bias, so are not good metrics to use. A drop in the incidence rate of advanced cancers is a better and more convincing metric to present.

Response: We agree completely with this. Reduction in stage 4 diagnoses is a stronger indicator that there will be a reduction in mortality. We have ensured that this point is covered in the discussion. No significant shift in stage 4 diagnoses have been seen in the program as a whole so far, but we have referenced a recent publication from a longer established centre where late-stage rates are compared in areas where screening was available with those that only later adopted screening. This showed a 25% reduction in stage IV lung cancer. We also note, in defence of the screening biases highlighted, that the early-stage rate is not excessive as has been found in some other programs in Eastern Asia. We hope to be able to report change in mortality in the next 1 or 2 years.

"Participation rates, which compare favorably to other international experience, are evidence of strong public engagement, ..."

Can the "participation rates" be neatly summarized in one or a few statistics? I found I had to go searching and digging through tables to get a sense of it.

Response: Thank you – coverage and uptake data has been highlighted in the results and discussion for ease of reference.

What about program sensitivity, specificity, false positive rates, interval cancer rates, harms done, and ... ?

There seem to be a lot of LCS program quality indicators that are not presented. They could be put into tables in supplement if space requires.

Response: The three- and twelve-month LDCT data provide an early indication of management of indeterminate findings (this has been clarified in figure 1 in response to an earlier point).

You have rightly noted that we have not presented important quality indicators. This is because the effectiveness standards (performance indicators) were only developed during the initial years of the program and are still at the stage of incorporating stakeholder feedback and defining thresholds. We have included this information in the discussion. We await data from research data sets such as DART (see above), as well other planned work nationally which will provide further insights into this data. We felt it important to not delay access to the data available thus far.

“Future longitudinal comparisons of socioeconomic group-stratified stage distributions between lung cancers diagnosed through LCS and those presenting clinically should help clarify whether stage shift is being achieved in all groups or a specific socioeconomic group.”

The discussion in the manuscript leaves the impression that determining successful screening will be done through evaluation of stage shift to early disease. As mentioned before this is a poor metric. See

Feng X, Zahed H, Onwuka J, Callister MEJ, Johansson M, Etzioni R, et al. Cancer Stage Compared With Mortality as End Points in Randomized Clinical Trials of Cancer Screening: A Systematic Review and Meta-Analysis. *JAMA*. 2024;331(22):1910–7.

Response: Thank you, we have addressed this specific point, citing a paper showing a reduction in late stage in the longest established areas and the paper you mention above. We felt that presenting the data we have, at this point was an important contribution to the real-world evidence.

“Lung screening must meet the standards set for screening in the NHS, with focus on; quality assurance and performance; better data systems bespoke to the programme; and a more engaged research community to the develop the future programme improvements.”

Awkward language/grammar.

Response: Thank you – reworded.

“A further limitation is that more detailed participant record-level data with cancer diagnoses linked from national registration data are only available for the initial phases of the programme.”

This seems like an important limitation and is hard to understand. Why cannot cancers in the Registry not consistently be linked to screened participants over time?

Response: We recognise that this has been one of our key challenges and have taken care to maximise the quality and interpretation of the data available in proportion to this limitation. We have reworded to ensure this is clear. We have two data sets available. The whole programme data are simpler to collect, so achievable by all clinical sites, summarising data in aggregate nationally. A more detailed dataset, including a ‘patient

level identifier' (NHS number) were required from the 'initial sites' who were resourced to provide this deeper data. Only in patients with NHS number, was it possible to perform linkage. Due to the scale of this programme and data resource available, this hybrid approach was taken.

Figure 1

From text

"Targeted Lung Health Check (TLHC) Programme"

"invited to a lung health check (LHC)"

Use of TLHC versus LHC suggests that they mean different things and the former is for screening? And the later is for health and risk assessment to see if screening should be offered.

If his interpretation is correct, then in Figure 1, the first box to branch off to the right says "Did not attend TLHC". This branching off appears to come to early because the individuals have not yet had a LHC and an offer to attend TLHC (screening).

There is one arrow leading to "Lung cancer diagnosed" coming from "At 12 months". Arrows should be coming from all three boxes in that row.

Response: This figure has been clarified and we hope the above comment and an earlier one are now addressed.

Figure 2. Some of the fonts are too small to be easily read. The data points on the figure lines can be enlarged, and number of decimals can be reduced.

Response: The figure has been updated

Table 1. Acronyms are not explained.

Response: Thank you – we have checked that the acronyms are expanded upon. These are the same as in earlier figures, except f/up (follow up – number of days observed for event).

Figure 4. "2) LDCT attendance of those who attended a LHC"

Should it not be "LDCT attendance of those who attended an LHC and were found to be at high risk and were offered lung screening". If you are just looking at screening attendance in all who attended LHC, you are conflating eligibility and attendance.

Response: We agree, this figure is showing both eligibility and attendance at LDCT following the offer. The wording has been altered. The majority of participants offered a LDCT attended a LDCT (90% for whole program data from figure 1).

We have updated figure 4 to show the ORs for the participants assessed as high risk and attendance for LDCT in those participants. We have updated the text to draw out the main observations.

Figure 5. Not only is it questionable practice to look at early-stage cancer as a metric because of potential biases, such as resulting from over-diagnoses, using proportion is additionally poor practice, because a change in any one level of stage will lead to change in proportions of the other levels, even when the incidence rates remain unchanged. Ideally, monitoring changes in incidence rates of stage IV cancer is preferred.

Response: Please refer to response above.

Figure 6 is a Table.

Response: Thank you – label and cross references corrected.

The Supplement needs a Table of Content and Headings.

Response: Thank you – edit made

Reviewer #3 (Remarks on code availability):

No statement about code was made or code presented. But the coding involved was only for simple descriptive statistics and logistic regression models using standard R packages. Not much to worry about here.

No statement was made about data availability.

Response: None required

1. Dettori, J.R., Norvell, D.C. & Chapman, J.R. The Sin of Missing Data: Is All Forgiven by Way of Imputation? *Global Spine J* **8**, 892-894 (2018).

LCS Nature Medicine Paper Peer review Responses (second review)

Editor comments and formatting:

The editorial team finds that the current version of the manuscript is hard to read and not accessible to a general medical audience beyond lung cancer specialists. Please address below the following points to improve clarity and accessibility to the general audience:

Thank you for your guidance, which we have included in full and we agree makes clear improvements to the manuscript readability and access for a non-specialist reader.

- The introduction benefits from providing background context into global and UK lung cancer burden, what is the current landscape of lung cancer screening, brief description of previous implementation projects, and what are the standards and challenges in implementing LCS. Only use a single paragraph to introduce the study, including the aims.

This has been addressed

- In the introduction, please remove the very specific descriptions of previous studies and recommendations.

This has been addressed

- The manuscript is very heavy with abbreviations- please avoid the use of unnecessary abbreviations (e.g. LCS, NHSE, TLHC, LHC).

Most abbreviations have been removed, except where very frequent e.g. LHC and LDCT

- Please remove editorialising terms/language throughout the manuscript when describing the study and findings- i.e. “first”, “novel”, “largest/most comprehensive”, “clearly” and “success”

This has been addressed

- The discussion needs substantive editing to improve accessibility to a general reader. Rather than going into details about specific results in the discussion, please provide more big picture discussions around the lessons learnt, considerations and implications for implementing cancer screening programs in other countries

This has been addressed with an extensive re-write to improve the readability.

- Please minimise the use of jargon throughout the manuscript, particularly the discussion

This has been addressed

Please address the following editorial and formatting comments:

- Please note that per Nature Medicine style, the abstract needs to be unstructured with a limit of no less than 200 words

We presume you mean no more than 200 words and have reduced the word count accordingly. The abstract is now compliant

- Please move the Methods to the back of the manuscript

Addressed

- You currently have 8 main display items, including 5 figures and 3 tables. Please merge or convert 2 of your main display items to extended display figures/tables (each of which must fit on a single page). The extended display figures need to be labelled as such in the main text. Please note that we can permit 6 main display items and 10 Extended Data display items. Please convert some of your tables to Supplementary Tables, of which you can have an unlimited number.

Addressed. Figures are now reduced to 3 main with 2 labelled as extended display figures. Previous figure 3 was deleted as data were included in figure 1. We now have 3 tables and one supplementary

- Please be aware that we are unable to accommodate supplementary text. Please integrate them into either the main text or the methods section. Please note that there is no limit for our Methods section as it is online only.

The supplementary methods are now included at the end of the methods section

- Please move the two supplementary figures as extended display figures and label them as such in the main text

Addressed

- Note all tables cannot have fills- they must be white fill with black text

Addressed

- Any references cited only in the methods needs to be included in a separate methods-only references section and should be numbered contiguously to the main reference list (i.e. number starts at XX following on from the numbering of the main reference list, not 1).

Addressed

- Please provide a data availability statement. We do not allow that “data available upon request”- please make the data publicly available or alternatively provide more information for readers on any restrictions accessing the data and how the data can be accessed upon application.

Provided – if required we can provide details of future data access request service.

- Please provide a code availability statement. We do not allow “code available upon request”. Please either deposit the code in a repository, such as GitHub and provide the appropriate details. Alternatively, please provide more information for readers on any restrictions to accessing the code and how the code can be accessed, including

how the data can be accessed upon application, the contact details and timelines.
Provided

- The Lung Cancer Screening Research Consortium Authorship must be organised exactly as do the title page of main authors and their affiliations. They cannot be presented as tables nor can they be supplementary tables.

Amended as requested. If necessary we would be content to recognise groups in “acknowledgements”

**Please note that joint authorship has adjusted to move Dr Arjun Nair to joint first position rather than joint senior. On review of the authorship instructions, this is a more accurate reflection of the work contributed according to our interpretation of Nature Medicine guidance.**

The article file must only contain these items in this order:

- Title
- Author List and affiliations
- Abstract
- Introduction
- Results (with Subheadings)
- Discussion
- Acknowledgements
- Author Contributions
- Competing Interests Statement
- References (for main text only)
- Figure legends (for main text only)
- Tables (note: tables should be pasted into Word files as editable tables, not as images)
- Methods
- Data Availability Statement
- Code Availability Statement
- Methods-only References

We believe the article is now compliant

In addition to addressing the remaining points from the reviewers, please edit your manuscript to comply with our formatting guidelines for Articles, which are:

* Abstract: 200 words, unreferenced.

* Main text: 4000 words with subheadings for the Introduction, Results and Discussion

* References: up to 60 in the main text + 20 methods-only references

* Display items: up to 6 main display items (inclusive of figures and tables) and up to 10 Extended Data display items (inclusive of figures and tables). Extended Data are an integral part of the paper and only data that directly contribute to the main message should be presented.

* Online Methods: no word limit; please provide the methods consolidated in a single section at the end of the main text document

Addressed

Reviewer #1:

This Reviewer congratulates the authors of the manuscript in their efforts to respond to the recommendations which I have made and also those from the other Reviewers. Practically all have been taken on board and incorporated into the text.

This manuscript now not only provides a comprehensive and detailed account of the UK's Implementation of a National Lung Cancer Screening Programme (NLCSP), over the last five years, but will also be a very useful document for international groups starting or implementing LCS programmes.

The Standard Protocol and the QAS addition to the supplementary section is also a bonus for the external readers.

One specific comment on re-reading the manuscript, I question the inclusion of Table 2, which provides information from the initial phase on Incidental Findings (IF). This table provides data on the 77,182 IF from 114,430 individuals who had a LDCT in the initial phase. However, this 114,430 only represents 14.6% of the total number of individuals (528,684) who underwent an initial LDCT in the reported programme. As this is such a small proportion of the total number of individuals recruited into the current programme, is it a true reflection of what is happening? Would it not be wise to hold back on this IF data for another publication when the data was much more mature ?

Response: Thank you for acknowledging the hard work of the team and the rigour of the work and data presented by a wide, equitable team.

Regarding the incidental findings, whilst we agree that the numbers only represent a small proportion of the total, this is still more than 4 times that of NLST. Thus we have left this table in the main text.

Reviewer #1 (Remarks on code availability):

Reviewed the information on https://github.com/craig-parylo/610_tlhc

The information provided would require an individual with the necessary computer /IT experience in order to answer the question - are the results reproducible.

Recommend that you have a Reviewer with these IT skills to answer this question.

Thank you we have not made changes

Reviewer #2:

The National Lung Cancer Screening Programme presents significant and valuable data regarding the implementation of lung cancer screening. This practice is gaining traction globally, particularly in European countries. The manuscript makes a noteworthy contribution to the body of evidence surrounding the implementation of lung cancer screening, the impact, but also the challenges in implementation.

The authors have made substantial revisions to the manuscript, providing the reader with a more detailed understanding based on the available data. As future data becomes accessible, we can anticipate additional evidence on relevant key performance indicators, although it is too early to draw conclusions at this stage. Thank you, we agree that ongoing analysis will provide better and more detailed evidence

Reviewer #3 (Remarks to the Author):

I have read the authors' responses to all reviewers' comments and the revised manuscript with changes made. The authors for the most part have responded adequately to the reviewers' comments. In several places where presentation had shortcomings, it was due to lack of required data at this time, which to an extent will be overcome in the future.

Thank you, we will ensure that future analyses address these deficiencies and indeed this is necessary to monitor the quality of the programme.

AUTHORS' RESPONSE to REVIEWER 2's COMMENT: "Cancer diagnoses were derived from the cancer registration dataset rather than an individual patient record." If this is the case, then how were interval cancers identified? I see from reading later on that data to calculate interval cancers accurately were not available.

This is a key metric that we will monitor once routine patient level data are recorded in mid 2026

"An independent review of the NHS in England in 2024, highlighted the improvement in early stage cancer detection "...likely to be in significant measure due to the Targeted Lung Health Check programme...".¹²

I'm glad there is a space between "in" and "significant".

An important point and perhaps a better form of words would have been "...in no small measure.."

"Cancer detection is expressed as a proportion of the prevalence round LDCTs (equivalent to 'per participant')."

I am fine with this presentation if the authors prefer it. However, there is no reason

why not to present it as per cent, that is per 100 individuals. Generally, people have a clear interpretation of percent, but possibly less so of a straight probability.

Thank you, we have not changed this as we prefer this format.

Small thing: There are many spacing errors where the authors have two and even three spaces between sentences. I presume that the journal will clean up the presentation.

This has been addressed

Reviewer #3 (Remarks on code availability):

Code was not presented. Although the datasets for England are understandable complex, the code to analyze the simple descriptive epidemiological statistics should be simple and easily figured out.

This has been provided as per earlier comment